



# A global assessment of nitrogen concentrations using spatiotemporal random forests

Razi Sheikholeslami[1,2], Jim W. Hall[1,2]

[1]School of Geography and the Environment, University of Oxford, Oxford, OX1 3QY, UK

[2]Environmental Change Institute, University of Oxford, Oxford, OX1 3QY, UK

*Correspondence to*: Razi Sheikholeslami (razi.sheikholeslami@ouce.ox.ac.uk)

**Abstract.** Anthropogenic nitrogen fluxes into surface freshwater bodies significantly impair water quality (WQ), pose serious health hazards, and create critical environmental threats. Quantification of the magnitude and impact of WQ issues requires identifying the key controls of nitrogen dynamics and assessing past and future patterns of global nitrogen flows. To

achieve this, we adopted a data-driven, machine learning approach to build a space-time random forest model for simulating nitrogen concentration in 115 major river basins of the world. The proposed random forest-based WQ model regressed the monthly measured nitrogen concentration collected at 718 river stations across the globe for the period of 1992–2010 onto a set of 17 predictor variables with a spatial resolution of 0.5-degree. The resulting model was validated with data from river basins outside the training dataset, and was used to predict nitrogen concentrations in all river basins globally, including

many with scarce or no observations. We predict that the regions with highest median nitrogen concentrations in their rivers (in 2010) were: United States, India, Pakistan, Bangladesh, China, and most of Europe. Furthermore, our results showed that the rate of increase between 1990s and 2000s was greatest in rivers located in eastern China, eastern and central parts of Canada, Baltic states, southern Finland, Pakistan, parts of Russia, mainland southeast Asia, and south-eastern Australia. We found that, globally, the most influential predictors of nitrogen concentrations are temporal: month of the year and

cumulative month count, reflecting the secular trend. Apart from temporal variables, cattle density, nitrogen fertilizer application, temperature, precipitation, and pig population are the most influential predictors of nitrogen pollution of the river systems. The proposed global WQ model will provide a new tool to explore agricultural and land management strategies designed to reduce nitrogen pollution in freshwater bodies at large spatial scales.

## 1 Introduction

### 1.1 Background: anthropogenic nitrogen enrichment in global waters

Water quality (WQ) management and pollution control are vital for achieving water security and attaining human wellbeing as reflected in the UN Sustainable Development Goals (SDG6: clean water and sanitation; UNGA, 2015). However, intensifying human-induced changes on Earth and the environment have imposed immense challenges in sustainable WQ management (van Vliet et al., 2017). During the past century, WQ has declined because of unregulated wastewater





discharge, livestock manure and fertilizer draining into catchments (and aquifers). In addition, extensive construction of dams, excessive extraction of groundwater, deforestation, and expanding agricultural land use have altered sedimentary processes, mobilization of salts, and nutrient export to river systems, all of which drive WQ deterioration and groundwater pollution in many parts of the world (Scanlon et al., 2007; Seitzinger et al., 2010; Struyf et al., 2010). Furthermore, climate change is expected to have detrimental impacts on WQ due to perturbations of precipitation and temperature patterns in
hydrological cycle (Lu et al., 2020; Yang et al., 2019).

In particular, the modification of global nitrogen, which is one of the relevant substances for WQ requirements, is significant. As shown by Green et al. (2008), from 1800 to mid-1990, nitrogen loading to land surface has increased twofold because of the accelerated anthropogenic activities. For example, the reactive nitrogen produced by human in 2010 was approximately four times more than reactive nitrogen created by natural processes (i.e., biological nitrogen fixation) (Fowler
et al., 2013; Vitousek et al., 2013). The increased nitrogen flows can be mostly attributed to farming practices. In the past half-century, the global demand for food has boosted the agricultural intensification and expansion, accompanied by use of fertilizer and animal manure for enhancing crop yields (Foley et al., 2005; Godfray et al., 2010). The excessive use of fertilizer and manure together with a relatively low nutrient use efficiency by crops have resulted in large losses of nitrogen. Zhang et al. (2017) estimated that total manure nitrogen production increased six times during 1860–2014 with an overall
significant increasing trend. Additionally, the application of nitrogen fertilizer accounts for more than half of the total direct input of nitrogen to cropland (Fowler et al., 2013), and is expected to triple by 2050 (Mogollón et al., 2018).

Nitrogen is an essential nutrient for growth and nourishment of all living organisms and is a key element of dietary proteins. Nevertheless, an overabundance of nitrogen in water can cause highly undesirable consequences for human health (Miller et al., 2021). There is evidence that high nitrogen levels in drinking water is a plausible risk factor for infant
methemoglobinemia (Greer and Shannon, 2005) and colon cancer (McElroy et al., 2008). Excessive nitrogen loading of rivers can also create adverse environmental effects on aquatic and terrestrial ecosystems through three biochemical mechanisms (Jones et al., 2014): eutrophication, acidification, and direct toxicity, which might lead to numerous problems, such as proliferation of harmful algal blooms, exacerbation of hypoxic zones, fish mortality, and loss of biodiversity (Turner et al., 2003; Diaz and Rosenberg, 2008; Clark and Tilman, 2008).

Curbing the aforementioned negative effects of excess levels of nitrogen underscores the need to develop effective WQ management and restoration practices. A successful management plan typically requires mathematical models to simulate and predict nutrient cycling through the hydrosphere. There is a large variety of models and modelling concepts which differ in terms of process complexity, process inclusivity, time scale, and spatial resolution of input data and simulations. These models try to provide better understanding of how multiple hydrologic, atmospheric, and anthropogenic factors and their
interactions control water and nutrient flows (Arheimer and Olsson, 2003; Wellen et al., 2015). Importantly, with a growing understanding that WQ problems are global and pervasive, several attempts have been made in the last two decades aimed at





introducing and improving large-scale models. These models, in principle, have been built to investigate the regional, continental, and global patterns of WQ by considering biogeochemical fluxes between the atmosphere, terrestrial, and aquatic ecosystems.

Bearing in mind that our imperfect knowledge of the real-world systems and the associated uncertainties impede fully capturing the nutrient dynamics (Sheikholeslami, 2019), large-scale models are indispensable tools for sustainable WQ management because (for more discussion see Kroeze et al. (2012) and Tang et al. (2019)):

1.     They can provide globally consistent assessments which are necessary for identifying global WQ trends and hotspot areas of water pollution.

2.     They can elucidate the interplays among various large-scale drivers of global nutrient cycling that are difficult to account for in smaller scales, such as international trade of food and animal feed.

3.     They can help extrapolate WQ estimates in regions where there is no or insufficient data for a detailed assessment.

4.     They can be used to examine the spatiotemporal trends of nutrient concentrations under a wide range of future global scenarios, such as hydrometeorological conditions, land use change, hydrology regulation, etc.

5.     They can be effectively coupled with other global models constructed for replicating economy, land use dynamics, climate, livestock population, crop growth, and other components related to WQ.

In line with this, the hydrology and WQ modelling community has become increasingly sophisticated in the use of large-scale models. Despite a considerable advancement in model development, practical limitations, such as data scarcity and computational costs, still preclude effective application of the existing models on global scale (Döll et al., 2008; Kauffeldt et
al., 2013). This calls for continued improvement of efficient large-scale models to deepen our knowledge of spatiotemporal variability of WQ indicators globally. In the next subsection, we provide an overview of recent large-scale modelling studies, with particular focus on data-driven and machine learning models. We also discuss their characteristics and identify shortcomings.

**1.2 Motivation: harnessing the power of machine learning for large-scale WQ modelling**

In the last two decades, Earth system sciences have seen substantial effort in the development of large-scale WQ models simulating the nutrient cycle on continental and global scales. The complexity of large-scale models varies based on their objectives and applications. These models broadly fall into two groups. Process-based mechanistic models historically evolved to incorporate known (basin/local-scale) processes of system behaviors, and their development has mainly focused on solving the conservation equations for mass, energy, momentum, and kinematics under certain simplified assumptions. In



contrast, data-driven models link WQ parameters (i.e., pollutants) to environmental and anthropogenic variables based on
       weaker physical constraints. Data-driven models are often constructed using either empirical/statistical relationships or
       machine learning techniques. Note that this classification is not exclusive. Indeed, some of the large-scale models integrate
       both process-based and empirical representations of physical mechanisms; see, for example, He et al., 2011a; Beusen et al.,
       2015; Jackson-Blake et al., 2017.

Although process-based models are the important tool for understanding physical mechanisms, their robustness and accuracy
       may suffer from our incomplete knowledge of the hydrogeochemical processes and physical properties of the system when
       upscaling. These models often have many parameters that need to be calibrated or estimated, which is sometimes
       troublesome due to wide ranges of parameters and complex interactions among them. Further, a limited number of
       observation sites, required to configure (i.e., initialize, parameterize, and calibrate) such models, restricts their usefulness.
Another critical limiting factor for the application of these models at large scales is computational demand. Their simulation
       time typically exceeds the computational resources available for a comprehensive analysis of the model behavior under
       different conditions. Examples of process-based models that have been previously applied to large-scale WQ modelling
       include INCA (Whitehead et al., 1998), SWIM (Huang et al., 2009), DLBRM (He and DeMarchi, 2010), HYPE (Donnelly et
       al., 2013), SWAT (Abbaspour et al., 2015), and VIC-RBM (Raptis et al., 2016).

On the other hand, data-driven machine learning (ML)-based models can learn nonlinear, non-monotonic relationships
       between system state variables without explicit mechanistic information on the processes. ML algorithms can model the
       spatiotemporal dynamics of a dependent variable (e.g., nitrogen concentration) as the function of a set of chosen predictor
       variables (e.g., precipitation and temperature values, fertilizer use, etc.), purely based on given data. More importantly, ML-
       based models can extract information from diverse datasets, possibly including those that are typically not used in process-
based models. In addition to solving prediction problems, ML algorithms are useful tools for performing classification and
       grouping of data as well as building rule-based systems. It is important to note that establishing a physics foundation for ML-
       based models can be achieved by supplying the physical knowledge to the model via a proper statistical analysis of the input-
       output data and the choice of adequate input variables (Guillemot et al., 2021; Guo et al., 2020), and applying physical
       constraints on input-output relationships by penalizing solutions that violate physical principles.

The opportunity presented by ML-based models is based on the fact that although observational data for global WQ
       indicators are scarce, data that measure the drivers of these parameters (i.e., covariates) are not. In fact, the explosion of large
       environmental datasets along with rapid advancements in artificial intelligence technology have caused ML methods to
       attain outstanding results in the regression estimation of WQ indicators at a range of spatiotemporal scales (Tiyasha et al.,
       2020; Sujay Raghavendra and Deka, 2014). Another important advantage of modern ML algorithms is their high efficiency
during the training procedure. Thus, ML-based models can offer valid and computationally frugal alternatives for projection
       of future change effects on surface WQ. Notwithstanding the success of ML-based WQ models, a few pitfalls have





hampered their wider adoption (Reichstein et al., 2019): first is the lack of interpretability of ML-based models. These models are often built for prediction and do not provide physical relationships, i.e., they are treated as "black boxes". Second, and more fundamentally, the ML-based estimations are typically prone to uncertainty because of the finite sample

size, i.e., not knowing the output variable at unsampled regions outside the training dataset. Third, choosing the best ML algorithm might not be easy due to the existence of many different algorithms. To further synthesize the literature from the last decade, Table 1 provides an overview of papers that have employed ML to simulate WQ at large spatial scales.

**Table 1. Recent studies published from 2011 to 2021 that applied ML to large-scale surface WQ modelling.**

| Reference | ML algorithm | Spatial scale | | Temporal scale | | WQ constituent(s) |
|---|---|---|---|---|---|---|
| | | *Resolution* | *Extent* | *Time step* | *Extent* | |
| He et al. (2011b) | ANN | River basins | Japan | Monthly | 1995 | TNC |
| Álvarez-Cabria et al. (2016) | RF | River basins | Spain | Annual | 2003-2009 | N, P |
| Ceccaroni et al. (2018) | DT | - | Wadden Sea | Daily | 2003-2015 | *Forel-Ule* index |
| Collins et al. (2019) | MLT | Lakes | United States | Seasonal | 1980-2011 | TN, TP |
| Ross and Stock (2019) | DT | - | Chesapeake Bay | Monthly | 1986-2007 | DO |
| Russ et al. (2019) | RF | 0.5-degree | Global | Annual | 1992-2013 | EC |
| Chen et al. (2020) | DT, RF, DCF | River basins | China | Weekly | 2012-2018 | pH, DO, CODMn, NH$_3$-N |
| Dony (2020) | RF | River basins | Great Lakes | Annual | 2000-2016 | DIN, SRP, PP |
| Mellios et al. (2020) | DT, SVM, RF | Lakes | Europe | Monthly | 1980-2009 | CBB |
| Shen et al. (2020) | RF | 30-arc-second | United States | Seasonal | 1994-2018 | N, P |
| Arias-Rodriguez et al. (2021) | ELM, SVR | Lakes | Mexico | Daily | 2013-2019 | Chl-a, TSM, SDD |
| Thorslund et al., 2021 | RF | River basins | Global | Monthly | 1980-2010 | EC |
| Wang et al. (2021) | RF | River basins | Texas Gulf Region | Seasonal | 2011 | NO$_3$−N, TP, *E.coli* |
| Zhi et al. (2021) | LSTM | River basins | United States | Monthly | 1980-2014 | DO |

From our review of literature, we make three critical observations:

1. Despite a plethora of powerful ML methods, ANN is the most popular method for WQ modelling. As reported by Tiyasha et al. (2020), from 2013 to 2019, the average number of published papers that used ANN-based WQ models was 20 paper per year and is still increasing.





2. Unlike the data driven empirical WQ models (e.g., Green et al., 2004; Schultz et al., 2006; Mayorga et al., 2010; McDowell et al., 2020), our thorough search of the relevant literature indicated that ML methods have rarely been implemented at global scale.

3. Despite being successful in simulating and predicting surface WQ at catchment-scale, ML methods have not been utilized to provide spatially explicit (gridded) estimates of WQ indicators. Based on our observation, almost all ML models are lumped in space (see Table 1).

The above-mentioned issues highlight the need for an improved ML-based model for efficient and effective estimation and mapping of the global WQ constituents at a higher spatial resolution. Such global model can further help assess the state of worldwide aquatic biodiversity, determine water-related health hazard over large areas, and evaluate impacts of global drivers such as climate change on WQ. This gap motivated our development of the spatiotemporal ML-based global WQ model in the present study.

**1.3 Objectives and outline**

The primary goal of this study is to introduce a global WQ model that is based on ML approach. The model is designed to estimate the distribution of nitrogen concentrations (nitrate-nitrite) across large global river basins for the period of 1992–2010 based on the observations recorded in the United Nations Global Freshwater Quality Database (GEMStat; https://gemstat.org/) and other environmental and anthropogenic variables. Our global model maps the predicted nitrogen levels at a 0.5-degree (≈ 55 km) spatial resolution and with a monthly time step.

To achieve this, we developed a spatiotemporal random forest WQ model. Random forest is a well-known ML technique that creates many decision trees from bootstrap samples of data. Random forest can efficiently handle large, complex, and heterogeneous geo-environmental datasets with a superior predictive performance (Fox et al., 2020; Knoll et al., 2019). To our knowledge, this study is one of the first attempts to estimate nitrogen concentrations at global scale using ML methods. Although we demonstrate the utility of the proposed random forest in simulating nitrogen levels, the overall modelling procedure presented here can also be implemented in conjunction with any ML technique for predicting other WQ indicators at large scales. Therefore, the main objectives of this research are:

1. To develop and validate a random forest-based predictive model for evaluating global levels of nitrogen using existing monitoring data.

2. To identify where the nitrogen pollution is most severe using predicted patterns of nitrogen concentration in large global river basins.

3. To determine key drivers of nitrogen variability at global scale.



The remainder of this paper will proceed as follows. Section 2 describes data gathering procedure and discusses our approach to selection of predictor variables. Details of model construction and the utilized validation strategy are presented
in Section 3. In Section 4, first, model performance evaluation and out-of-sample validation results are analysed, then we illustrate our model's usefulness for understanding the state of the WQ issue, i.e., nitrogen pollution hotspots and its causes. Finally, conclusions and recommendations for future research are given in Section 5.

## 2 Data collection and pre-processing

### 2.1 Global nitrate-nitrite measurements

For training our ML-based global WQ model and analysing its performance, we focused on nitrate-nitrite nitrogen (NOx—N) as the response variable in this study. NOx—N is one of the dominant forms of nitrogen and is very soluble in water, which can significantly deteriorate the quality of surface water. Nitrogen in synthetic fertilizer, manure, and wastewater can be decomposed to ammonia, which is then oxidized to NOx—N and will subsequently enter the groundwater, streams, and lakes, leading to eutrophication, hypoxia, or human health implications. We collected NOx—N
monitoring data from the GEMStat repository which was established by the UNEP GEMS Water Programme (Barker et al., 2007). GEMStat provides an online, globally harmonized, open-access database for WQ at global, regional, and local scales. It currently contains more than 3.5 million observations for rivers, lakes, reservoirs, wetlands, and groundwater systems from approximately 3,000 stations. Overall, data is available from 1965 to 2017 and for approximately 250 WQ indicators.

NOx—N is well represented in the data repository of GEMStat, with more observations and fewer missing values (da Costa
Silva and Dubé, 2013). In total, GEMStat records 82,302 NOx—N observations from 718 stations located in 75 countries (see Fig. 1). For our modelling purpose, we extracted data with the length of 18 years across all river monitoring stations. This resulted in 42,413 temporally consistent NOx—N observations within the 1992–2010 period, from which we computed the monthly aggregated values. The selected period was also compatible with most of our predictor variable datasets (see section 2.2).




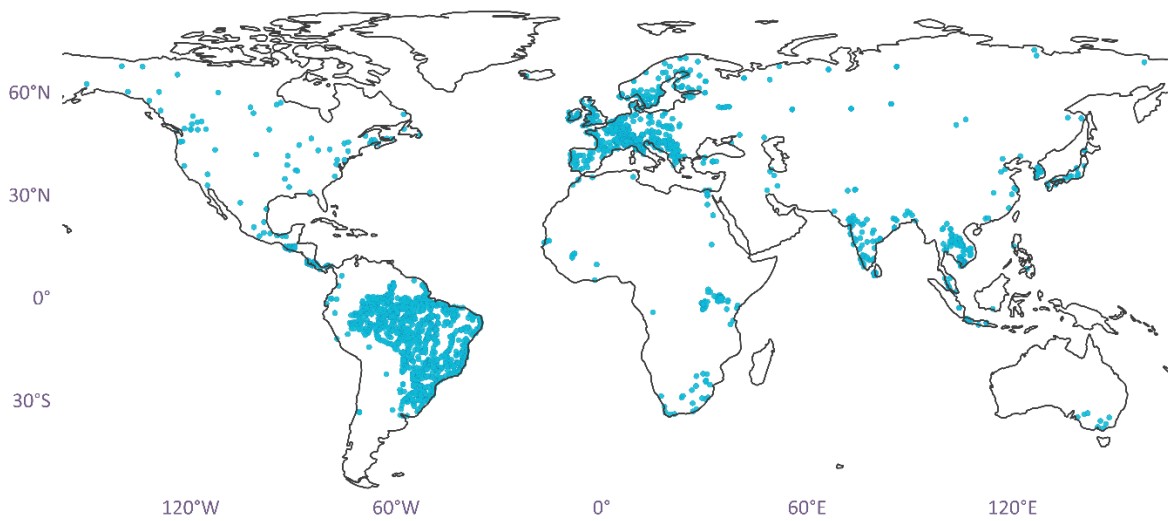

**Figure 1: Spatial distribution of GEMStat (https://gemstat.org/) monitoring stations (blue dots) used in this study for building ML-based WQ model.**

Fig. 2 displays the distribution (PDF) of the global NOx—N values. Note that 10% of the monthly observations in our training dataset have values smaller than 0.02 mg/L (10th percentile) and 10% of them have values larger than 1.46 mg/L (90th percentile). Also, about 7.24% of the GEMStat-monthly NOx—N measurements are less than 0.001 mg/L. The mean and standard deviation of the monthly measurements are 0.63 and 2.23 mg/L, respectively. To improve the data symmetry and suitability for use in our ML-based model, all observed NOx—N concentrations were transformed using Box–Cox 195 technique. The optimal Box–Cox transformation parameter was obtained using the maximum-likelihood approach.





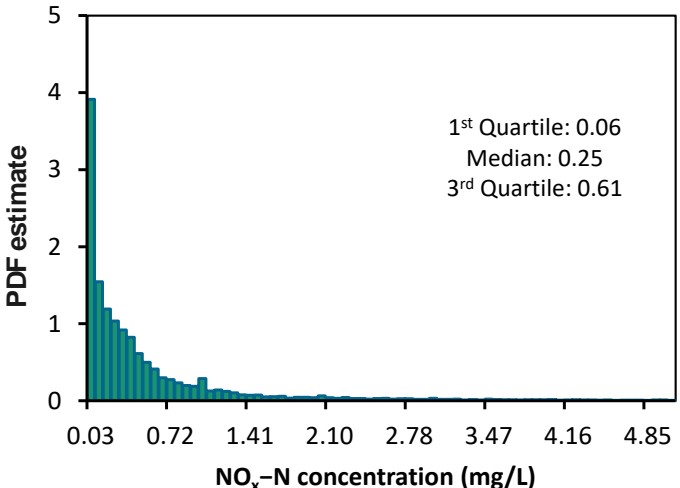

**Figure 2: Probability density function (PDF) estimate of the global (monthly aggregated) NOx—N measurements across all GEMStat monitoring stations, over the 1992-2010 period. Note that the sum of the bar areas is less than or equal to 1.**

**2.2 Predictive variables selection**

Several processes interacting at different spatiotemporal scales, along with varying intensities, drive variability of the nitrogen in water. Thus, the primary concern of using any ML method for global modelling is selecting an optimal combination of the predictor variables. In this study, we systematically identified predictors considering three selection criteria, including process representation, model complexity, and data availability, as follows.

Using domain knowledge obtained from consultation with experienced modelers, we first determined an initial set of variables likely to control nitrogen concentration. As a result, a total of 27 potential explanatory variables were identified. Then, we screened the list of variables to reduce the risk of including a bevy of redundant information and generated a more parsimonious selection. This has been achieved in a process-informed manner through an extensive literature review (for more discussion see Lintern et al., 2018; Billen et al., 2013; Bouwman et al., 2013 and the references therein). Lastly, among

these widely known controls on constituent concentrations, we chose a subset of relevant and discriminative predictors based on the open data criterion. Therefore, when data were not readily available in existing resources, the variables were excluded from the list. As can be seen from Table 2, the selected space-time predictors can be generally categorized into three classes of anthropogenic, hydroclimatic, and geographic factors.

**Table 2 An overview of predictor variables, their spatiotemporal scales, and sources used in this study**

| Variable | Spatial scale | | Temporal scale | | Source |
|---|---|---|---|---|---|
| | *Resolution* | *Extent* | *Time step* | *Extent* | |
| Livestock population (Sheep, chicken, pig, cattle, buffalo) | 5-arcmin | Global | - | 2010 | Gilbert et al. (2018) |





| Human population | 0.5-degree | Global | Annual | 1990-2015 | Kummu et al. (2018) |
| --- | --- | --- | --- | --- | --- |
| Wastewater production | 5-arcmin | Global | - | 2015 | Jones et al. (2021) |
| Cropland area | 0.5-degree | Global | Annual | 1961-2014 | Jackson et al. (2019) |
| Synthetic nitrogen fertilizer use | 0.5-degree | Global | Annual | 1900-2013 | Lu and Tian (2017) |
| Precipitation | 0.5-degree | Global | Monthly | 1900-2013 | Willmott, and Matsuura (2001) |
| Temperature | 0.5-degree | Global | Monthly | 1900-2014 | Willmott, and Matsuura (2001) |
| Runoff | 0.5-degree | Global | Monthly | 1902-2014 | Ghiggi et al. (2019) |
| DEM | 15-arc-second | Global | - | 2010 | Danielson, and Gesch (2011) |


For these datasets, all coordinates were projected into WGS84 global reference system. Then, for each predictor, the *k*-nearest neighbour imputation was applied to replace the missing values with the mean values obtained from the 100 nearest neighbours found in the data via a Euclidean distance metric. Because of the different resolutions, we subsequently reshaped all predictors to the same spatial resolution of 0.5-degree using bilinear interpolation resampling technique and the same 220 temporal interval. After standardizing data (i.e., cantering at the mean and scaling by the standard deviation), all predictor variables were Box-Cox transformed. Finally, the 13 selected predictors were matched to the monthly NOx—N concentrations of monitoring stations shown in Figure 1. This resulted in a training dataset containing a total of 34,115 matched monthly measurements and all predictors. In this way, we produced a consistent, spatially explicit global-coverage datasets for the years 1992–2010.

**3 Model development**

The assembled dataset is represented by a collection of observations $\mathbf{Y} = \{y(s_l, t_\tau), \ (s,t) \in \mathcal{S} \times \mathcal{T} \subseteq \mathbb{R}^2 \times \mathbb{R}\}$ measured at $l = 1, 2, \dots, n$ spatial locations and at $\tau = 1, 2, \dots, T$ time points over spatial domain $\mathcal{S}$ and temporal domain $\mathcal{T}$. In the present study, $y(s, t)$ can be considered as a realization of water quality process, i.e., NOx—N concentrations. There are various data-driven approaches for analysing and automatically extracting information from Y without the need to construct explicit 230 physical models. Compared to traditional data-driven methods such as Kriging, inverse distance weighting, and nearest neighbor interpolation, ML methods are increasingly becoming more popular mainly because they cope better with high-dimensional data and are not restricted to linear relations (Domingos, 2012). For this study, as described in the next sub-sections, we designed a space-time ML-based model using the random forest method.

**3.1 Proposed spatiotemporal random forest model**

Random forest is a relatively modern ML method that basically uses the assemblage of multiple iterations of decision trees. With the capability of processing large environmental datasets and handling nonlinear relationships, random forest has been



increasingly become a popular data analysis method that outperforms other ML-based tools (Fox et al., 2020). Details of random forest algorithm can be found in Breiman (2001); Liaw and Wiener (2002); Friedman et al. (2009) and were briefly described below.

In summary, random forest is nonparametric regression technique which contains a collection, or "forest," of independent regression trees $\{t^{(k)}: k = 1,2, ..., B\}$ as base learners. Each growing tree, $t^{(k)}$, in the forest is made from bootstrap samples drawn, with replacement, from the original training dataset. These trees are formed by randomly selecting $m$ variables out of $p$ predictors at each parent node, and the best split-point is found among these m variables using greedy recursive algorithm to create two child nodes. This greedy algorithm recursively partitions a group of $m$ predictor variables based on identifying

the predictor that minimizes error when regressed against the output of interest. Note that trees are grown deep with no pruning.

In the random forest algorithm, the remaining observations, which are not included in the bootstrap sample, are called Out-Of-Bag (OOB) sample (also referred to as test set). For each tree the prediction performance (measured, for example, by mean squared error (MSE)) on the OOB sample is recorded and is used for measuring the prediction error of the $k$-th

regression tree. After calculating all individual tree predictions, they are averaged to obtain the final random forest prediction. This process works as a cross-validation for each tree in the forest and provides an unbiased overall model error estimate (Prasad et al., 2006). Hence, the prediction at a new site, $\hat{f}_\varphi$, with predictor vector $\boldsymbol{x}$, is found by estimating the mean value of all regression trees, $\hat{f}^{(k)}$, i.e.,

$$\hat{f}_\varphi(\boldsymbol{x}) = \frac{1}{B}\sum_{k=1}^{B} \hat{f}^{(k)}(\boldsymbol{x}; \theta_k) \, , \tag{1}$$

where the variable $\theta_k$ determines which predictors get included in the $k$-th tree. **Eq. (1)** explains the main idea behind random forest, which is averaging over $B$ fitted regression trees to reduce variance, and thus to improve predictive performance compared to a single regression tree.

### 3.1.2 Measuring variable importance using random forest

An important built-in feature of the random forest is its ability to assess the variables' predicting strength (expressed as

variable importance ranking) using the recorded OOB prediction errors. This feature helps screen the relatively small number of important factors from the pool of selected predictor variables, thereby identifying which variables are strongly driving the WQ indicator of interest. Random forest evaluates variable importance by estimating the mean decrease in prediction accuracy before and after randomly permuting the values of a given predictor in the OOB data. For the $k$-th tree, when randomly permuting the $i$-th predictor, MSE of the OOB data can be calculated as (Wei et al., 2015):

$$MSE^{(k)} = \frac{1}{N_{OOB}}\sum_{j=1}^{N_{OOB}} \left(y_j^{(k)} - \hat{f}_j^{(k)}\right) \quad \text{and} \quad MSE^{(k)}{}_i = \frac{1}{N_{OOB}}\sum_{j=1}^{N_{OOB}} \left(y_j^{(k)} - \hat{f}_{j,i}{}^{(k)}\right) , \tag{2}$$





where $N_{OOB}$ is the OOB sample size; $y_j^{(k)}$ is the $j$-th observation in the OOB data of the $k$-th tree; and $\hat{f}_j^{(k)}$ and $\hat{f}_{j,i}^{(k)}$ are predictions of the OOB data before and after randomly permuting $i$-th predictor, respectively. Note that, in **Eq. (2)**, if the $i$-th predictor is not selected on the split-point of any node of the $k$-th tree, then then $\hat{f}_j^{(k)} = \hat{f}_{j,i}^{(k)}$ (for all $j$), and thus $MSE^{(k)} = MSE^{(k)}{}_i$.

Assuming that permuting the values of one predictor cannot increase prediction errors if that predictor dose not significantly impact model accuracy, the difference between $MSE^{(k)}$ and $MSE^{(k)}{}_i$ is then averaged over all trees and considered as importance measure. In other words, the overall importance of $i$-th predictor, $PIM_i$, can be expressed as the mean decrease in accuracy values of all trees:

$$PIM_i = \frac{1}{B} \sum_{k=1}^{B} \left( MSE^{(k)}{}_i - MSE^{(k)} \right),\tag{3}$$

The worse the model performs when the $i$-th predictor variable is randomized (i.e, higher $PIM_i$), the more important that variable is in terms of predicting the response variable.

### 3.1.3 Incorporating space and time into standard random forest

The standard random forest method does not exploit the spatial and temporal information of the observations, essentially being '*aspatial*' and '*nontemporal*' algorithm. When used for modelling spatiotemporal data, the standard method generates

a single output which is estimated from the whole extent of the study area, using all available data points over time. For WQ modelling, however, this can be a crucial problem since WQ constituents are naturally characterized by spatial and temporal heterogeneity, which indicates that the true underlying relationship between predictant and predictor variables can be spatially and temporally varying. In the present work, therefore, it is necessary to construct a random forest model that can adequately capture the spatial-temporal characteristics of the nitrogen levels.

Two strategies have been proposed in the literature to account for spatial information. The first strategy uses a hybrid modelling framework by embedding Kriging and Gaussian process modelling into the standard random forest method (Saha et al., 2021; Canion et al., 2019). The second strategy is more straightforward because it explicitly utilizes geographic information as the auxiliary inputs, for example, by adding geographic coordinates (Behrens et al., 2018; Meng et al., 2018) or other spatial distances (Li et al., 2011; Wei et al., 2019) into the list of predictors. We followed the second strategy by

incorporating latitude and longitude into our WQ model as two additional predictors since they contain geographical information. Regarding the time dimension, we added two more time-variables, namely Cumulative Month since 1992 (CM) and Month of the Year (MOY) to represent distance in the time domain, thereby better capturing dynamics of nitrogen levels.

Next, we produced a global space-time regression matrix for our model by binding these auxiliary variables and values of

predictors together. As a result, the final set of predictors consists of 17 variables which are expected to be key determinants





of source, mobilization, or delivery of nitrogen concentrations globally. Fitting a random forest model for this space-time regression matrix follows the same procedure as described previously in section 3.1.1. Based on the fitted model, we generated timeseries of NOx—N concentrations as raster maps within the space-time domain of interest. Fig. 3 is the schematic illustration of how the proposed spatiotemporal random forest can be applied to predict NOx—N at a new

location.

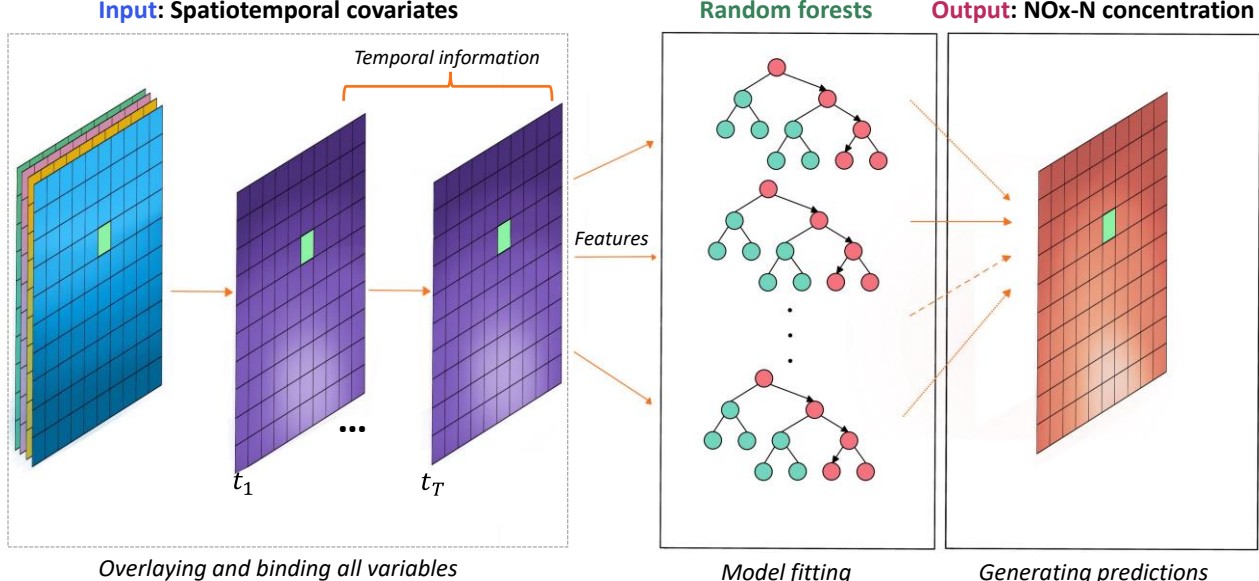

**Figure 3: A schematic representation of the proposed spatiotemporal random forest model.**

**3.2 Model training and tuning**

When training ML methods, the performance of algorithm is highly sensitive to how the dataset is partitioned into training and testing samples. To tackle this issue, we implemented k-fold cross validation strategy. This validation strategy starts by randomly splitting dataset into k subsets of similar size. Then, a random forest is learned using observations in $k - 1$ subsets, and an error value is calculated by testing the algorithm on the remaining set. The *k*-fold cross validation estimation of the error is the average value of the errors committed in each fold. Additionally, the process of partitioning can be repeated

several times (known as *repeated k*-fold cross validation) to create multiple random splits of the dataset. It has been often presumed that performing repeated *k*-fold cross validation on different random partitions can stabilize the error estimation, leading to the reduction in the variance of the estimator (Rodriguez et al., 2010). In the present work, we conducted 10-fold cross validation with 3 repeats. The overall cross validation accuracy was taken as the average of mean absolute error (MAE) obtained from each repeat.





Unlike many data-driven algorithms, there are only two tuning parameters need to be calibrated for random forest: (i) the number of variables selected randomly at each node ($m$) and (ii) the number of trees in the forest ($B$). For a random forest model, $B$ controls the variation among different regression trees and m determines the extent of overfitting. Increasing the number of trees typically decreases the prediction error of the random forest up to a certain point. For $B$ values larger than this threshold, model accuracy changes very slightly, whereas computational demand increases significantly (Liaw and

Wiener, 2002; Freeman et al., 2015).

## 4 Results and discussion

### 4.1 Model performance assessment

Our global random forest model was tuned by varying $B$ between 300, 400, 500, 800, and 1000, and m between 2, 4, 8, 12, and 14. Based on these experiments, the optimal setting was found to be $B = 500$ and $m = 2$. For the final model, the 10-

fold cross-validation (repeated 3 times) yielded average MAE value of 0.44, suggesting a good performance of the model. Fig. 4(a) further verifies the effectiveness of the model by comparing response values estimated by the trained model with the observed values (in a Box-Cox transformed scale), across all monitoring sites. As additional performance criteria, we calculated the coefficient of determination (($R^2$) and Nash-Sutcliffe efficiency criterion ($e_{NS}$) between all cross-validated predictions and their corresponding observations. In Fig. 4(a), we can see that the observations lie reasonably close to the

predicted concentrations, with $e_{NS} = 0.86$. The obtained $R^2$ value implies that the proposed random forest algorithm has accounted for 81% of the variability of the observed NOx—N values. This confirms the ability of the random forest model in accurately establishing the relationships between monthly NOx—N observations and predictor variables. Note that, for poorly predicted NOx—N values, the observations are, in general, larger than predicted values, indicating conservative estimations by the proposed model. Attending to this issue is particularly important when applying our model to evaluate

risk of high nitrogen concentrations.

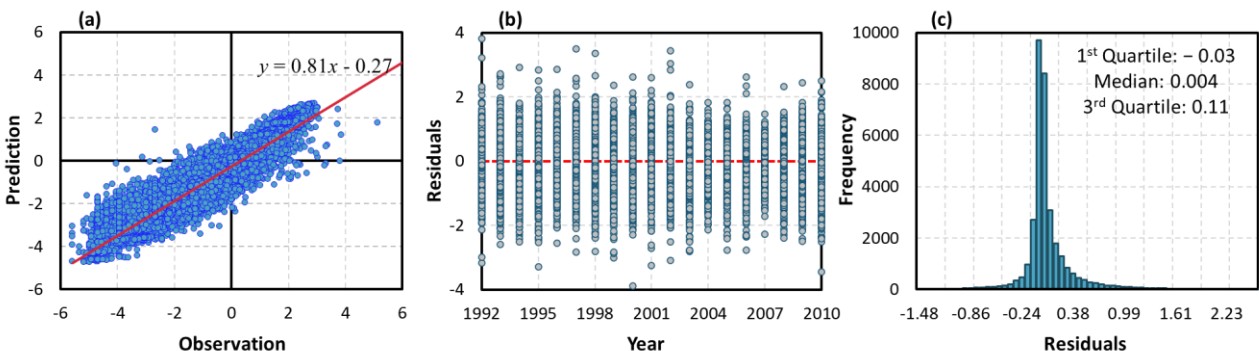

**Figure 4: Performance of the random forest model for predicting NOx—N concentrations. Subplot (a) shows the pairwise scatter plot of the monthly average observations versus corresponding cross-validated predictions. The solid red line represents the linear**





**regression fit. Subplot (b) shows temporal distribution (annual average) of the residuals and subplot (c) depicts histogram of the residuals, obtained from all training stations. Note that subplots (a) and (b) are in Box–Cox-transformed space, whereas subplot (c) is in the original scale (back transformed).**

We performed a thorough residual analysis in Fig. 4(b) and (c). The temporal trend of residuals for all monitoring stations (in a Box-Cox transformed scale) are displayed in Fig. 4(b). As can be seen, the average annual residuals are approximately

cantered around zero and are relatively constant over time and space. This confirms that our model can capture the spatiotemporal structure in the NOx—N data. Fig. 4(c) depicts the histogram of the residuals obtained from all simulated and observed (back transformed) data, which has mean of 0.10 mg/L and standard deviation of 1.93 mg/L. We observe that the distribution of the residuals is close to being approximately normally distributed, though it has heavier tail than a normal distribution.

Also, we carried out an out-of-sample validation where a portion of the data which was not used to build the model served as an independent test set for out-of-sample estimation of the performance of the proposed model. Recall that, to train the random forest model, we used monthly NOx—N observations which cover 1992 to 2010 (sample size = 34,117). The rest of the observations (sample size = 28,802) which does not belong to this period was considered as '*new*' data for out-of-sample testing. The scatterplot in Fig. 5(a) shows the out-of-sample validation results for NOx—N estimates. We see that global

monthly NOx—N estimates are well correlated with ground measurements, with $e_{NS}$ and $R^2$ values of 0.60 and 0.61, respectively. Hence, the proposed model reproduced the *new* NOx—N values with a reasonable accuracy. This fair agreement between random forest predictions and independent NOx—N observations provides confidence to the overall approach. The histogram of the residuals obtained from all simulated and observed out-of-sample data is shown in Fig. 5(b), with mean of -0.51 mg/L and standard deviation of 4.10 mg/L.


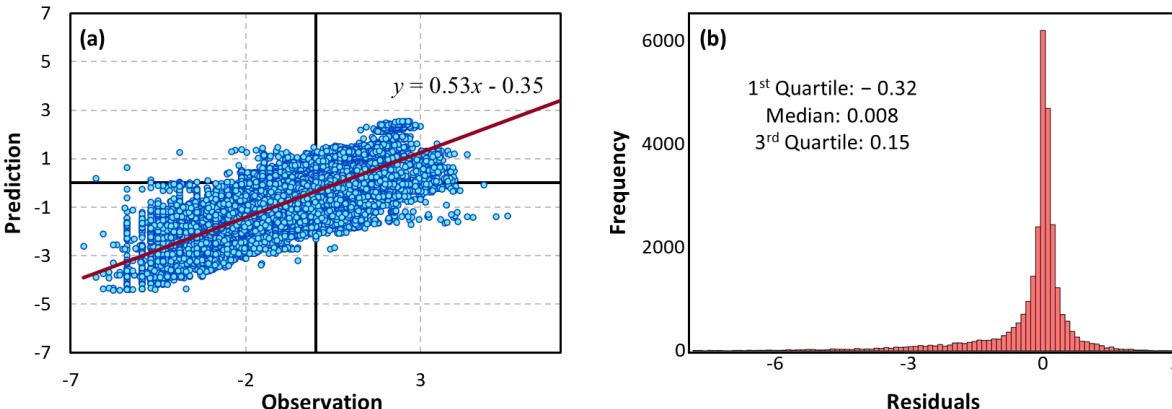

**Figure 5: Out-of-sample performance-estimation of the random forest model for predicting out-of-sample NOx—N values. Subplot (a) shows the pairwise scatter plot of the monthly average observations versus corresponding predictions in out-of-sample set. The solid red line represents the linear regression fit. Subplot (b) depicts histogram of the residuals, obtained from all out-of-sample**

**data. Note that subplot (a) is in Box–Cox-transformed space, while subplot (b) is in the original scale (back transformed).**



## 4.2 Patterns of nitrogen concentrations and global hotspots

To characterize temporal and spatial variations of nitrogen, we generated global spatial maps for each month using our random forest model for every year in the period of 1992–2010. Since large drainage basins have historically played a vital role in the localization of cities as well as the distribution of human activities, our temporal and spatial analysis was cantered on the world's 115 major river basins. In this paper, the basin polygons (in black) were derived from the Global Runoff Data Centre (GRDC, 2020). The total area of this river basins covers ~70% of the global land surface area with ~60% of the global population. Fig. 6 presents the estimated spatial distributions of mean NOx—N across 115 major river basins averaged over 1992-2010. As can be seen, the estimated NOx—N concentrations exhibited a considerable spatial variability over the globe. The highest rates of nitrogen can be especially found in many European basins, United States, parts of Mexico, southern Brazil, eastern Argentina, West Africa, South Asia, eastern China, parts of South Korea and Japan.

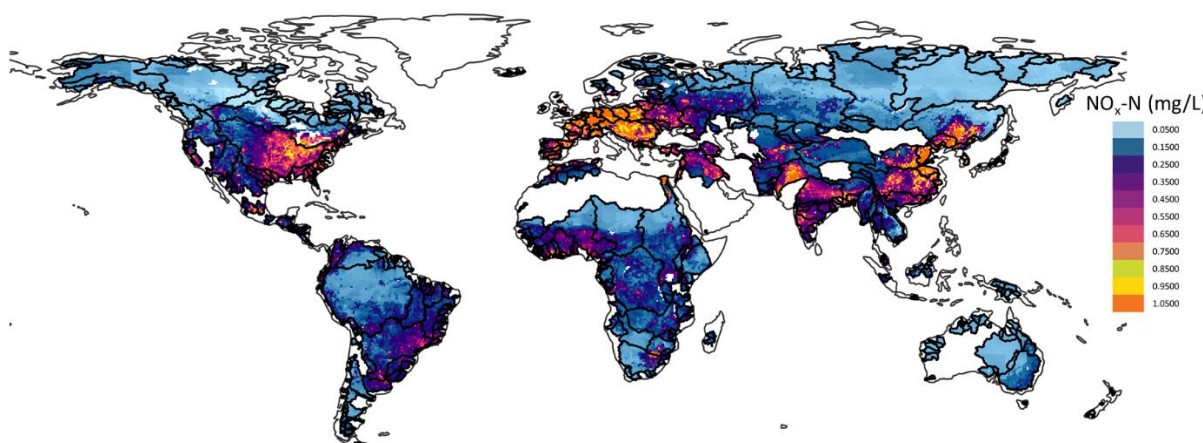

**Figure 6: Simulated global map of NOx—N concentrations averaged over 1992–2010, across major river basins of the world.**

To investigate the spatial pattern of nitrogen changes from the year 1992 through 2010, we averaged the estimated NOx—N concentration in each grid cell during the 1990s (1992–1999) and 2000s (2000–2010). Fig. 7 depicts the relative percentage change between these two periods. This map can reflect the spatial heterogeneity of NOx—N concentration change over study period, avoiding extreme values in a specific year. As shown in Fig. 7, almost entire eastern China has experienced an enhancement of NOx—N concentration. In addition, eastern and central parts of Canada, parts of South America, southern France, Switzerland, parts of Balkans, Belarus, Baltic states (Latvia, Lithuania, and Estonia), southern Finland, Pakistan, Afghanistan, parts of Russia, mainland southeast Asia (Cambodia, Laos, Burma, Thailand, and Vietnam), and south-eastern Australia showed a significant increasing gradient of NOx—N concentration in their rivers from 1990s to 2000s (over 20% at some locations). However, model results also indicate that in many regions of the world there has been a considerable decline in nitrogen levels (at some locations more than 20% decrease) during the past decades, including south Korea, India, Ukraine, Poland, Germany, United Kingdom, Central Africa, and northern Brazil.






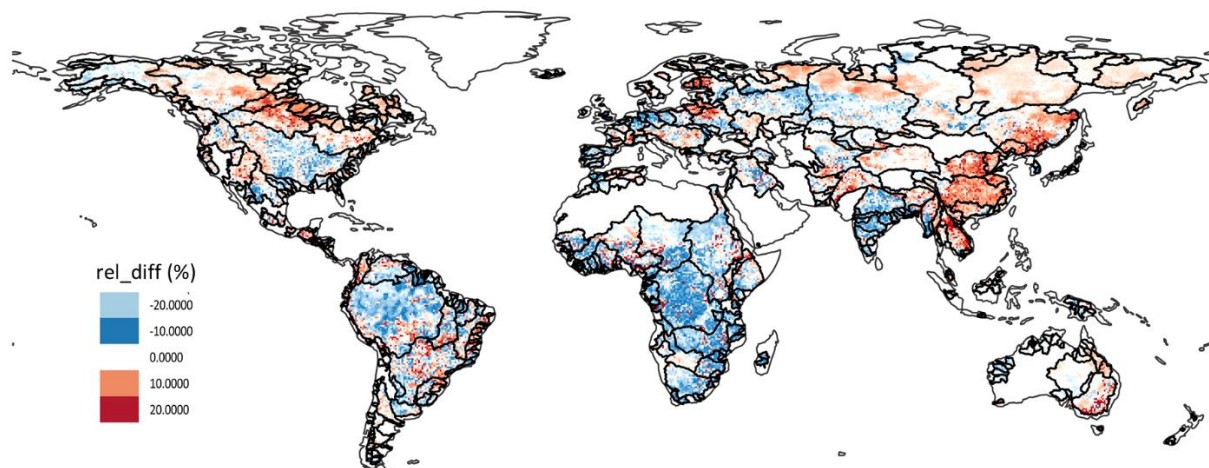

**Figure 7: Spatial pattern of NOx—N concentrations change during the 1990s (1992–1999) and recent decade (2000–2010).**

The latitudinal distribution of average (annual) estimated NOx—N concentrations during 1992-2010 is shown in Fig. 8. We clearly observe that the NOx—N concentrations were low for areas with low temperature and little precipitation. In contrast,
the dominant regions of the concentrations can be found in the Northern Hemisphere, where maximum values occurred in 20°N–60°N. These latitudes correspond to the high agricultural activity and high livestock densities (Potter and Ramankutty, 2009). In the Southern Hemisphere, the concentrations are generally much lower, except for 20°S–40°S, possibly due to high fertilizer use by Brazil, Argentina, and South Africa (Lu and Tian; 2017).

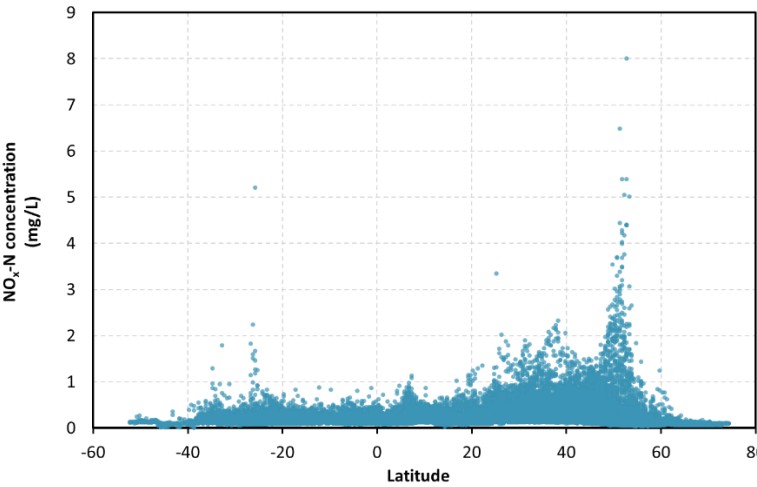


**Figure 8: The latitudinal distribution of predicted NOx—N concentrations obtained from Fig. 7. Each blue dot represents the value of each 0.5° resolution grid cell.**





By combining the results shown in Fig. 6-8 we identified several river basins over the globe where NOx—N was densely concentrated, and accordingly the nitrogen pollution might be most serious. Based on our results, the high nitrogen level has
occurred in basins with extensive agriculture and low precipitation surpluses, which was also reported by previous studies (see, e.g., Shindo et al., 2003; He et al., 2011a). Overall, major hotspots of NOx—N were the river basins of the Mississippi, Sebou, Egypt's Nile Delta, Indus, Gang, Yellow, Yangtze, Yongding He, Huai He, Nakdong, Kitakami, Lower Amur, and the Lake Urmia basin as well as most of the European river basins, such as Rhine, Danube, Vistula, Thames, Trent, and Severn.

**4.3 High importance factors influencing predictions of nitrogen levels**

For the proposed random forest model, we examined predictor variable importance in Fig. 9. The relative importance of the 17 selected variables to predicting nitrogen levels was measured by mean decrease accuracy criterion (as described in section 3.1.2). As shown in Fig. 9, two variables representing temporal dynamics of nitrogen (MOY and CM) are the most influential variables. This reveals that, distinctly, the most important factor for predicting monthly NOx—N concentrations is
time, i.e., cumulative and/or month of the year. In other words, incorporating these variables into the random forest allows our model to fit different spatial patterns for each month. Other strongly influential predictors are (in rank order): (i) cattle population, (ii) nitrogen fertilizer use, (iii) temperature, (iv) precipitation, and (v) pig population. We also observe that elevation (DEM) proved less important to overall model accuracy. Presumably, the effect of elevation was already accounted for by the other spatial covariates (i.e., latitude and longitude).


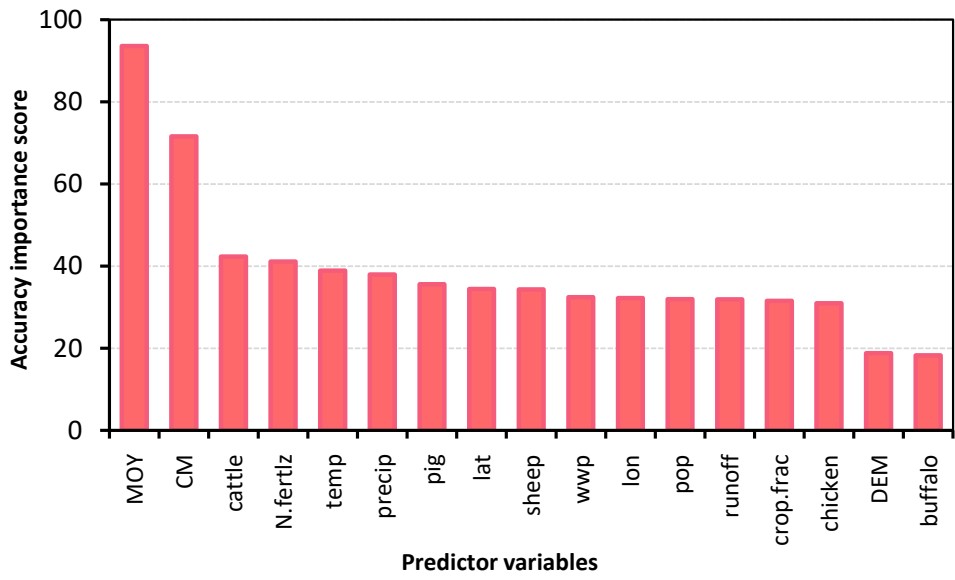

**Figure 9: Importance plot derived from the proposed random forest model. The horizontal axis lists predictor variables in order of decreasing importance. The vertical axis (unitless) represents the mean decrease in accuracy divided by the standard deviation of**





**the decrease in accuracy after permuting the variable of interest. Note that only the relative importance scores between predictors**
**should be interpreted, not the absolute values of the scores on y-axis.**

As demonstrated in Fig. 9, among different livestock categories used in this study, the number of cattle and pigs contributed the most to model accuracy, followed by sheep and chicken densities, while buffalo population is a non-influential variable mainly because buffaloes are likely less common than other species. It is worth mentioning that, based on FAO (2018)'s estimates, more than 50% of the global manure-nitrogen input (both manure applied to soil and manure left on pasture) was
produced by cattle in 2005. Moreover, stocks of pigs increased by ~140% worldwide, which caused more than five-fold increases in the nitrogen applied to soils from pig manure during 1961-2014 in Africa and Asia (FAO; 2018). This together with our modelling results highlight that livestock activities (e.g., manure/slurry application, animal housing, milk parlour washings, etc.) is one of the primary accelerators of water contamination in global scale.

The role of nitrogen fertilizer use as a chief determinant of nitrogen level is also reflected in Fig. 9 wherein nitrogen fertilizer
use is among the top four strongly influential predictors. The high importance of this variable can be justified because of both cropland expansion and raised fertilizer application rate in per unit cropland area globally. This supports the results presented in our hotspot analysis as well (see Fig. 6). Based on our analysis, some of the river basins, where nitrogen pollution is most severe, are in the top five fertilizer-consuming countries (China, India, the US, Brazil, and Pakistan), which accounted for more than 60% of global fertilizer consumption (Lu and Tian, 2017). We also found that wastewater
production is a moderately influential predictor for the NOx—N concentrations, clearly because it is not the largest source of nitrogen contamination of surface water in global scale (Billen et al., 2013). However, the disposal of wastewater with a low level of treatment might have considerable implications for managing anthropogenic nitrogen flows in highly populated river basins, as the point sources of nitrogen are primarily associated with wastewater drainage.

Although stream WQ is significantly related to runoff from agricultural areas and synthetic fertilizer application during
agricultural production, the fraction of cropland area does not seem to have a significant contribution to the model performance (see Fig. 9). It is very surprising to see such a low importance score for cropland area. This might be partially due to a high correlation between agricultural fraction of land area and nitrogen fertilizer use. In fact, the impact of cropland area might be accounted for by the other correlated variable (i.e., nitrogen fertilizer use). Strobl et al. (2007) reported that when using permutation-based mean decrease in prediction accuracy as an importance measure, there might be a bias in
estimating importance of correlated variables by random forest algorithm. Another possible reason for this observation might be related to the nature and scale of the predictor variable itself. Pham et al. (2021) asserted that permutation-based metrics can underestimate importance of zero-inflated variables that are heavily skewed (e.g., cropland area) compared to variables that are more normally distributed (e.g., temperature).

As illustrated in Fig. 9, model performance was also highly sensitive to both temperature and precipitation, though
temperature tends to be slightly more important. The high importance of these hydroclimatic variables implies that climate





change can strongly control nitrogen contamination across the globe. For example, there is evidence that high temperatures (e.g., summer heatwaves) promote blooms of harmful cyanobacteria, through increased biological growth rate when nitrogen is present (Myer et al., 2020). However, evidence for the impact of precipitation on WQ is mixed, depending on the timing, magnitude, and recurrence of precipitation, though random forest should be capable of capturing these non-linear effects. It

is generally hypothesized that more extreme precipitation events generate greater nutrient runoff (Lu et al., 2020). On the other hand, nutrient concentrations can be reduced through greater flushing due to precipitation (Ho and Michalak, 2019). It is noteworthy that, with climate change expected to perturb temperature and precipitation regimes in most regions across the world, climate change impacts on WQ have often been overshadowed by water quantity-related problems (i.e., droughts and floods).

## 5 Conclusions and future research

The high-volume and intensive crop and livestock production permitted by widespread fertilizer use has led to substantial change in nitrogen loads in rivers globally. Given the anticipated increase in human and livestock population, reactive nitrogen is a major environmental threat not only to water bodies but the air and soils with repercussions for human health and biodiversity. Tackling this problem requires understanding of the trends in global nitrogen cycle. Our knowledge of

global nitrogen cycle has deepened rapidly over the last decades with the aid of new measurement technologies and advanced mathematical models. Yet, the salient processes are extremely complex and process-based models have struggled to fully represent the spatial variability that is evident in measured nitrogen concentrations.

To address this issue, we proposed a dynamic ML-based WQ model that uses random forest algorithm to simulate nitrogen levels (i.e., nitrate-nitrite concentrations) at large spatial scales. Random forest estimates nonlinear regression functions

using an ensemble of non-smooth, data-adaptive family of learners known as regression trees. Our proposed random forest-based WQ model offers two notable advantages:

1. First, it is a data-driven approach. As such, it estimates nitrogen concentration directly from data by exploiting the random forest's ability of finding complex spatiotemporal patterns (without providing explicit form of them). This makes our model easier to construct in comparison to process-based models.

2. Second, unlike process-based models that are computationally expensive in nature and often suffer from over-parameterization and calibration issues, our model is computationally efficient and is less sensitive to different parameter settings, so it can function quickly over large datasets.

In this study, we assembled several site-level measurements and a comprehensive set of readily available gridded environmental data to build a random forest-based predictive model. The predictor variables used in this paper represent



various anthropogenic, hydroclimatic, and geographic factors. The algorithm was trained using repeated 10-fold cross validation strategy and the results indicated that the proposed model has a high predictive performance. We then predicted monthly nitrogen levels worldwide (180°E–180°W; 90°S–90°N) with a spatial resolution of 0.5° × 0.5° (55 km × 55 km at the equator) and a time span from 1992 to 2010. The produced maps (in GeoTIFF format) were also used to characterize global WQ trends and hotspot areas of nitrogen concentration across 115 major river basins of the world. Furthermore, we
performed variable importance analysis to determine the contribution of each predictor variable to model performance. Consequently, we identified a subset of important predictors (and the associated processes) for the nitrogen concentrations in global scale. Overall, based on the numerical results presented in this paper, our main findings are as follows:

1.   On average, in United States, Europe, India, Pakistan, and China nitrogen pressures have occurred over large areas during 1992-2010, where severe water pollution occurs, depending on climate, drainage networks, and other factors.

2.   We observed an increasing gradient of nitrogen concentration change from 1990s to 2000s in China, eastern and central parts of Canada, western United States, Baltic states, southern Finland, Pakistan, parts of Russia, mainland southeast Asia, and south-eastern Australia.

3.   Variable importance analysis revealed the prominent role of nitrogen fertilizer use and livestock density in nitrogen pollution of the river systems. Thus, implementing technical measures for improving crop-livestock farming
practices must be at the forefront in reducing nitrogen environmental losses.

4.   The predicted nitrogen levels also showed a significant sensitivity to hydroclimatic variables (i.e., temperature and precipitation), which will be of growing concern in the context of global climate change.

For future research, the spatial and temporal results on WQ derived from our proposed model can be used by ecological, hydrological, and human health models as well as by decision makers in two important directions. First, they can be applied
in the context of scenario analysis to explore the nitrogen concentration's sensitivity to a wide range of plausible future changes. Second, they are particularly useful for assessing the impacts of nitrogen enrichment on public health, water resources, and biodiversity. Also, future work may include employing advanced global sensitivity and uncertainty analysis techniques for uncertainty apportionment. As suggested by Sheikholeslami and Razavi (2020), the given-data techniques are well-suited for this purpose when dealing with data-driven models.

Nevertheless, we do not advocate ceasing development of physical process-based models. Depending on the scale of spatial and temporal inference, we encourage use of hybrid models and model ensembles that integrate ML methods and process-based approach (see, e.g., Corzo et al., 2009; Kraft et al., 2020). Furthermore, there are ample possibilities for development of physics-informed (or physics-constrained) ML-based WQ models to learn a wide range of physics relevant to hydrologic processes (Nearing et al., 2020). For example, an intriguing, simple strategy is that the performance metric used to train ML-

based models can be modified to account for the physical consistency of the model predictions. Finally, it should be noted that, given the flexibility of the random forest algorithm, this paper does not aim to draw any conclusive remarks on the extrapolation capability of the random forest-based models.

**Code Availability**

The modelling was performed in R statistical computing environment, which provides multiple open-source packages for
ML. The results obtained in R can be easily converted into any of the standard georeferenced formats to produce digital images or maps afterwards. Our code uses a set of fast and scalable R packages, such as "randomForest", "sp", "caret", "ranger", and "CAST", which facilitate automated raster-based workflows. The modelling experiment was run on an Intel(R) Core(TM) i5-4670 CPU @ 3.40 GHz computer and 32.0 GB of RAM. The parallel processing was enabled using the "doParallel" package in R. In the spirit of reproducible research, the full procedure of the proposed global model, starting
from the data collection and pre-processing to the 0.5-degree raster predictions will be available on GitHub upon publication of the manuscript (https://github.com/Razi-Sheikh/GLOBAL_WQ).

**Author contributions**

All authors contributed to conceiving the ideas of the study and designing the method and experiments. RS collected and compiled datasets and prepared the R codes for the spatiotemporal random forest algorithm. All the simulations have been
carried out by RS. RS wrote the manuscript with contributions from JWH. All authors contributed to the interpretation of the results, structuring, and editing of the paper at all stages.

**Competing interests**

The authors declare that they have no conflict of interest.

**Acknowledgements**

This paper was written during the COVID-19 pandemic. We would like to take this opportunity to acknowledge a deep sense of gratitude to the healthcare workers for leading the battle against coronavirus pandemic around the world. This research was funded by the Wellcome Trust, Our Planet Our Health (Livestock, Environment and People - LEAP), award number 205212/Z/16/Z.



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
