# Peer review of "A global assessment of nitrogen concentrations using spatiotemporal random forests"

_Hydrology and Earth System Sciences, 2021_

## Referee Comment (RC3)

**Review** of manuscript "A global assessment of nitrogen concentrations using spatiotemporal random forests" by Razi Sheikholeslami and Jim W. Hall

Sheikholeslami and Hall, 2022 utilize machine learning approach to build a random forest model of nitrogen concentration in major river basins. They apply the model to global river basins to identify nitrogen concentration hot-spots and decadal increase in nitrogen concentration. The manuscript applies random forest approach for estimating riverine nitrogen concentration, but their findings are not filling a gap in the literature. Their finding that highest nitrogen concentrations are observed in United States, Europe, China, and India is not new given that these regions are agriculture dominated and utilize large amount of fertilizer and manure. The manuscript lacks a novel motivation, several important predictor variables are not considered, and the final analysis does not provide any new information.

**Introduction:**

1. The authors include "three critical observations" from the literature review of ML applications for studying water quality. What is the reason for including these observations?

2. The two main objectives of the study related to identifying global nitrogen pollution hotspots and key drivers of nitrogen variability at a global scale are weak since various studies have examined these and it is not a novelty.

3. The Introduction currently has three section and is quite long. Condensing this section will improve the readability of the manuscript.

**Data and Methodology:**

4. Lines 218-222: The authors mention reshaping all predictors to 0.5-degree resolution, but they have not described how precipitation, runoff, and other predictors from the entire upstream catchment/watershed is accounted for in predicting river nitrogen concentration. It appears that they only considered the contribution of the various predictors from the 0.5-degree grid. It is highly likely that only a fraction of this grid falls in the upstream catchment and for several

observations large fraction of the upstream catchment is not included in this grid. Thus, the estimation of contribution from various predictor variables is likely flawed.

5. Adding latitude and longitude as predictor variables does not fully capture the spatial relationship between observations and predictors.

6. Additional land use predictors should be considered in the study for example, forest fraction, urban fraction.

7. In addition to considering monthly precipitation, extreme precipitation variables and variables capturing dry spells should also be considered as they impact nitrogen concentration. For example, a long dry spell will result in high nitrogen concentrations however, if the dry spell is followed by a large precipitation event concentrations will drop. The monthly mean precipitation will ignore this temporal variability and thus not accurately predict nitrogen concentration.

8. Table 2 should be replaced with a table containing the final 17 predictor variables. In its current format the table 2 does not list the four time and space predictors and lists the livestock predictors in a single row that makes it appear as a single predictor not five different predictors.

**Results:**

9. In the Results section, the authors have not compared their findings with any of studies listed in Table 1. The authors should discuss the similarity and differences between their findings and that of others using similar model building approach.

10. In addition to the annual concentrations (Figure 6), authors should also analyze and discuss monthly or seasonal maximum concentration.

11. The authors had developed spatial plots of nitrogen concentration for every month between 1992-2000, then why did the limit the analysis of change in nitrogen concentration to a decadal scale difference only. For large river basins, they can perform trend analysis. This will me more useful information for policy makers than decadal scale difference.

12. Line 387 – 389: What factors contributed to the decline in nitrogen concentrations? Just merrily stating decline is not enough and the drivers behind this observation should be discussed especially given that few regions with highest nitrogen concentration (India and South Korea) also have the largest decline.

13. The fact that month of year (MOY) is significantly more important than precipitation, runoff, and temperature seems concerning. If precipitation, runoff, and/or temperature, were selected as predictor variables it would have a direct physical meaning. For example, increase/decrease in precipitation can decrease/increase the nitrogen concentration.

14. What is the physical meaning behind cumulative month (CM) variable being identified as the second most influential predictor?

15. Figure 9 - Why is the relative importance of 3-15 predictors almost same?

16. Did the most influential predictors vary over space and/or time?

---

## Author Comment (AC1)

**Response to Reviewers' Comments**

This document contains copies of all comments of the reviewers (*in italicized, blue* text) and our planned effort to address them (in normal, black text). Our proposed manuscript revisions are underlined.

**Reviewer 1**

*Overall I was not impressed by the methodological advancements or the scientific implications of this project. The authors built a standard ML algorithm, random forests, and applied it to a global nitrogen dataset. The portrayal of it as a spatiotemporal model is misleading: It is a standard random forest that uses month and lat/lon as additional predictors, which is perfectly fine but not a novel type of random forest model.*

We are thankful to the reviewer for the time and effort spent on reviewing our paper. We believe that these comments help us reduce the possible confusion in the text with respect to paper's novelty and rigor that appear to have arisen due to lack of sufficient clarity in the original version.

We regret that the novelties and motivations of this study were not clearly stated in the original paper. To address this comment, we will revise the Introduction section. More importantly, to improve the linkage between study objectives and results, we propose to present additional results to highlight several model capabilities. These include model validation using a new validation strategy, adding partial dependence plots, presenting the distribution of R2 values, adding new predictor variables (i.e., forest fraction, urban fraction, and hydrography data) and further discussions on limitations of our approach along with recommendations for future studies (more details can be found in our response to the reviewer's comments throughout this document).

Regarding the use of lat/lon, as we extensively discussed in Section 3.1.1, two strategies have been proposed in the literature to account for spatial information: (1) using a hybrid modelling framework by embedding Kriging and Gaussian process modelling into the standard random forest method (Saha et al., 2021; Canion et al., 2019); and (2) using geographic information as the auxiliary inputs, for example, adding geographic coordinates (Behrens et al., 2018; Meng et al., 2018) or other spatial distances (Li et al., 2011; Wei et al., 2019) into the list of predictors. Since adding lat/lon as predictor variables to represent spatial relationship is common in data-driven modelling of Earth and environmental systems, we followed this strategy and incorporated lat/lon into our random forest model. During the revision, we will highlight the weakness of the second strategy in not fully capturing the spatial relationships between observations and predictors.

Overall, we believe that the reviewer has under-estimated the contribution in the paper. As outlined in this rebuttal document, the application of the random forest model can be considered to be state-of-the-art in a rapidly evolving field. The use of these techniques in the very challenging field of global water quality modelling is novel and has yielded predictive results that exceed other approaches in important respects.

*The model appears to perform well on held out data, so the authors take that as evidence to then apply it globally and make maps. From those maps and the model itself, the authors pull out generally banal conclusions that have been recorded elsewhere. The 'so what' of the paper from the discussion section is that it could conceivably be used by other stakeholders in applications, but this link felt very light, so I am left skeptical of any pathway to impact.*

We regret that the reviewer regards the conclusions as being banal. We admit that in trying to summarize a complex global picture we did not manage to fully highlight the originality and significance of the results.

The purpose of the model is to enable water quality assessment and quantification of future scenarios (e.g., of livestock pasture extent) at a global scale. Though some instances of analysis of these questions exist (e.g., Mayorga et al., 2010; He et al., 2011; Beusen et al., 2016; etc.) they all have inevitable limitations, and none have used our proposed approach which we believe yields worthwhile results, so it is important that the methodology is peer reviewed in the hydrological literature.

We further examine model validation by implementing another validation strategy known as Leave-Location-and-Time-Out Cross-Validation (LLTOCV) during the revision. For more details, please see our response to your comment regarding extrapolation issue. In addition, to address the comment on the innovations that this study brings, we will revise the Introduction and relevant sections in the Discussion to include a more substantive 'so what' from the paper.

**Specific feedback:**

*The literature review of ML methods in water quality (Section 1.2) does not powerfully motivate the present study. Why are the "three critical observations" listed interesting and relevant? This section overall feels disconnected from the rest of the paper.*

We remain convinced that our 'three critical observations' from review of the literature were strong enough to motivate this study because: first, despite a plethora of powerful machine learning methods, ANN is the most popular method for water quality modelling. To fill this gap, we applied random forest algorithm to examine its predictive performance when applied to large spatial scales. Second, our thorough search of the relevant literature indicated that machine

learning methods, particularly random forests, have not been implemented at global scale to identify hotspots of nitrogen concentrations nor to systematically determine high importance factors influencing nitrogen variability. Third, despite being successful in simulating and predicting surface water quality at catchment-scale, machine learning methods have not been utilized to provide spatially explicit (gridded) estimates of nitrogen levels. Based on our observation, almost all machine learning models are lumped in space (see Table 1). This gap further motivated our development of the spatiotemporal random forest-based global model in the present study. In the revised manuscript, we will improve the writing to reduce the reviewer's confusion that appear to have arisen due to lack of sufficient clarity in the original versions.

*The authors also fail to discuss why so few papers attempt to apply their models at a global scale (extrapolation risks) and make it seem that the community just never tried before, which is not the case. Broadly, the motivation for this work is not clear and compelling.*

Thanks for pointing this out. In the revised manuscript, we will explain these major challenges associate with building data-driven models at global scale, particularly extrapolation risks and sampling bias. It is worth noting that we believe that by addressing these obstacles we have made a significant methodological contribution.

*"The primary goal of this study is to introduce a global WQ model that is based on ML approach. (Section 1.3)". There are hundreds of water quality models that are based on ML, as they mention in their previous section.*

As mentioned in Section 1.2 of the paper, we agree with the reviewer that there are many ML-based water quality models at catchment scale, but there are only a few models at '**global scale**'. In the revised manuscript, we will further clarify our research goal, which is the use of an appropriately refined ML method to predict spatial and temporal variations in WQ at a global scale.

*This paper is not really about providing a new dataset then, but a new model. **But their model is relatively off the shelf and does not tell us anything about the systems we do not already know.** Overall building a ML model, because we can, is not a compelling motivation in 2022.*

We disagree with this remark at several levels. There is a long tradition of publishing the application of mathematical and statistical methods to hydrological problems. Because of the complexity of hydrology (and water quality in particular) this is far from trivial. The application

was definitely not "off the shelf", involving careful and rigorous selection, testing and adaptation of the ML method, alongside a vast exercise in data preparation.

Regarding the lack of new information, we politely disagree. We'd like to highlight that our model was validated with data from river basins outside the training dataset and was used to predict nitrogen concentrations in large river basins globally, providing new information about dynamics of nitrogen concentrations in location with scarce/no observations. Furthermore, we provide NOx-N concentrations (mg/l) worldwide (180°E–180°W; 90°S–90°N) at a spatial grain of 0.5-degree. The NOx-N concentrations, mapped across the globe for 1992-2010, are available in a compressed GeoTiff file format in the WGS84 coordinate reference system (EPSG:4326 code). The developed stream nitrogen concentration maps have a wide array of potential applications in stream ecology, biodiversity research, conservation science, and stream and lake restoration ecology. For instance, the produced maps can be used to quantify the overall mass of nitrogen discharged into a specific lake or ocean body, enabling a deeper understanding of global-scale eutrophication. Furthermore, our estimates of nitrogen concentration can be used to verify new process-based models that predict nitrogen concentrations and transformations in inland waters worldwide. We encourage potential users of the described geo-dataset to contact the authors for future product updates. We will add this to the revised manuscript to better highlight the usefulness of our findings.

**Observational data:**
*Why is the data collection period ceased in 2010? Data continues to be collected, so this seems an arbitrary cutoff removing potentially more data.*

Data collection was not an arbitrary process in this study. We agree that new water quality data is collected on a continuous basis. As mentioned in the paper (Section 2.1), the observational data used for this study was obtained from the GEMStat repository which provides an online, globally harmonized, open-access database for approximately 250 water quality indicators. Based on our analysis, we found that the spatial coverage of the monitoring stations for the 1992-2010 period was sufficiently high to train our model, whilst for other time periods the number of monitoring stations were smaller.

Furthermore, unlike other temporal ranges, the quality of dataset within the 1992–2010 period was higher, in terms of temporal consistency and number of missing values. The selected period was also compatible with most of our predictor variable datasets (see Section 2.2). In addition, as mentioned in Section 4.1 (lines 351-354), the rest of the observations (sample size = 28,802) which does not belong to the 1992-2010 period was used for out-of-sample testing. To address

this comment and improve the clarification of data collection, we will add explanations in Section 2.1 on why this time span was selected in our study.

*Second, the stations used to build the model have large geographic disparities the authors do not discuss at length (e.g. abundance of sites in Brazil and Europe).* **Sampling bias by location is a huge consideration when applying the map globally.**

Thank you, this is a very good point. The sampling bias is a common challenge in machine learning-based water quality studies. To address this comment, we will introduce new results that map R2 values spatially. We will add a new 'Discussion' section to better describe caveats/limitations of our approach and will reflect on this issue (i.e., sampling bias), how it was corrected for and how it might, nonetheless, impact our results.

*I think this manuscript is an unsupported (by the data validation presented) extrapolation of a model to locations far different that those used to train the model. The authors gloss over this critical consideration when making the main global maps (Figs. 6-8).*

The performance of the model in spatial extrapolation has been extensively tested. Our results in Fig. 5 clearly shows that the proposed model reproduced the *new* NOx—N values at sites outside the training dataset with a reasonable accuracy. This fair agreement between random forest predictions and independent NOx—N observations provides confidence to the overall approach.

In the revised manuscript, we further investigate the extrapolation issue to address this very important comment. We will implement another validation strategy known as Leave-Location-and-Time-Out Cross-Validation (LLTOCV). In this method, like standard cross validation (CV), the dataset is split into folds again, but this time each fold left the data of complete locations or time steps or locations as well as time steps out. Over-fitting of the model in space and time can be then quantified by comparing the random 10-fold CV results with this 'target-oriented' validation results. Consequently, the high difference between 10-fold CV implemented in this study (lower error estimates) and LLTOCV (higher error estimates) can be an indication of spatial over-fitting as the models can very well predict on subsets of the time series of the locations used for training but fail in the prediction of unknown locations. We will also expand the relevant discussions in Sections 3.2 & 4.1.

Predictors data:

*The authors say they started with a list of 27 candidate variables but then reduced it by more than half, to around 13 variables, to "reduce…redundant information" but one of the key advantages of random forests is that they work well with highly correlated variables. What were the other variables considered that were ultimately not included?*

We politely disagree the comment that '*one of the key advantages of random forests is that they work well with highly correlated variables*.' In fact, the effect of correlations on random forest algorithm has been studied by many researchers, particularly how it might impact variable importance measured by random forest (see, e.g., Archer and Kimes, 2008; Strobl et al., 2008; Nicodemus and Malley, 2009; Nicodemus, 2011; Auret and Aldrich, 2011; Toloşi and Lengauer, 2011; etc.). For example, Archer and Kimes (2008) observed that the Gini measure of random forest is less able to detect the most relevant variables when the correlation increases, and they mentioned that the same is true for the permutation-based importance measure. Auret and Aldrich (2011) also confirmed these observations, and Toloşi and Lengauer (2011) called it "correlation bias". In summary, most of these studies reported two key impacts of the correlation on permutation importance measures: (1) the importance values of the most discriminant correlated variables are not necessarily higher than a less discriminant one, and (2) the permutation importance measure depends on the size of the correlated groups. In an important study, Gregorutti et al. (2017) provided theoretical validations for these assertations, in a particular statistical framework.

Since one of the main objectives of our study was identifying key drivers of nitrogen variability at global scale, we tried to reduce the number of correlated predictors to minimize the negative impact of correlations. To address this comment, we will provide the list of 27 potential explanatory variables in Appendix of the revised manuscript. We will also add relevant discussions on the correlation and variable importance in random forests in Section 3.1.2 and provide more appropriate references.

*Were the datasets aggregated over the watershed boundaries corresponding to each sampling location (for variables like precipitation and runoff they need to be)?*

Thanks for raising this point. Please note that we modeled nitrogen concentrations at 0.5-degree gird cell (not at watershed scale). Thus, the current predictors for in-stream nitrogen concentrations prediction only cover the properties within the grid cell of interest.

This can be resolved, for example, by considering hydrography data delineating global river networks, though it will presumably add more complexity to the model. In the revision, we will add more variables to the list of predictors, including upstream characteristics, stream proximity (e.g., distance up to the stream) or log-transformed flow accumulation for better capturing

spatial characteristics of watersheds. Previous studies have shown that these variables can be key drivers of water quality responses in rivers (see, e.g., Staponites et al., 2017; Lintern et al., 2017; Grabowski et al., 2016; Peterson et al., 2010). Particularly, they reported that accounting for the hydrological flow paths and flow accumulation through the landscape and coupling these processes with specific landscape features can improve model performance. We will also elaborate more on this with the supporting literature.

*Land cover is also known to be relevant but only cropland area was included.*

This is a valid concern. Note that, however, the predictor selection process used in this study were based on an extensive literature review and domain knowledge. In fact, the random forest-based model presented in this study were developed using those key controls previously identified, without any additional predictor selection processes. We found that, among various land related variables, the cropland area has been frequently reported as the factor that strongly influences nitrogen variability. Our numerical results also confirmed this fact as shown in Fig. 9. To address this comment, we will add more land-related variables (e.g., forest fraction and urban fraction) to the list of predictors and will investigate how it may impact model performance and results.

**Model development:**
***The novelty of this random forest methodology is greatly overemphasized.*** *There is research into spatio-temporal random forests, but those are far more advanced than what was applied here, making the title of the paper misleading. Here is an off-the-shelf random forest that anyone taking a Coursera data science course could apply successfully. To clarify, I am okay with the algorithm but troubled by the emphasis on its importance and novelty. Including latitude and longitude as predictors hardly makes this a spatial statistical model. Including month of the year hardly makes this a time series model.*

Thanks for your opinion, please see our response to the first two comments of this document. We also did not want by any means to imply that the idea of using lat/lon or cumulative month or month of the year were the novelty of this paper. In fact, the term '**novel**' has not been used even once throughout the paper, and we have not suggested that the algorithm is novel. So, it is not clear to us why the reviewer concluded that '***The novelty of this random forest methodology is greatly overemphasized***'. Furthermore, we'd be more than happy if the reviewer helps us identify some of the studies '*into spatio-temporal random forests, but those are far more advanced than what was applied here*' in the context of water quality, so we can cite them in the revised manuscript to strengthen the literature review part.

Regarding the merit of the utilized methodology, we restate the points about our study that we have made above, which the reviewer does not seem to have recognized or has under-estimated. We simply do not think it is accurate to dismiss a comprehensive exercise in data processing, model selection, adaptation, and validation as something "*that anyone taking a Coursera data science course could apply successfully*".

*Testing set: I would like to see sites completely held out as well to see how well the model predicts at new locations.*

As mentioned in our response to the second comment regarding extrapolation issue, we are currently running Leave-Time-and-Location-Out Cross-Validation (LLTOCV) strategy, and we will update the relevant results and discussions in the revised manuscript. In this method, like standard cross validation, the dataset is split into folds again, but this time each fold left the data of complete locations or time steps or locations as well as time steps (LLTO) out.

Model evaluation:
*What is the distribution of R2 values by location? Presumably some locations perform better than others.*

Thank you for this recommendation. We agree with the reviewer that some locations certainly perform better than others in terms of prediction accuracy. This is mainly due to the lack of sufficient, well-distributed measurements. In the revised manuscript, we will add a figure to show how R2 values vary in space. This figure will further help us evaluate the performance of the model.

*Also, the metrics are produced on a log-transformed scale. **What is the mean absolute error or root-mean-squared error in interpretable, mg/L units? A strong performing model in log-log space is quite easy to produce (across domains, not just water quality) so it is important to record performance metrics in the back-transformed data space relevant to decision makers.***

We are rather surprised by this comment. The reviewer certainly knows that data transformation is a common procedure in building empirical/machine learning-based models. In practice, it has been suggested that when implementing supervised learning algorithms, training data and testing data need to be transformed in the same way. However, there is a misconception that data transformation is not necessary for random forest.

To better explain this issue, first, note that random forest is a tree-based model and tree-based models do not care about the absolute value that a feature takes. They only care about the order of the values. Second, note that there is a distinction between the output of the random forest when it is used in classification or regression problems. For classification tasks, the output of the random forest is the class selected by most trees. For regression tasks, the mean or average prediction of the individual trees is returned. By taking these into account, we can say that data normalization, therefore, won't affect the output for random forest classifiers, whereas it will affect the output for random forest regressors. Regarding the regressor, it can be shown that the algorithm will be more affected by the high-end values if the data is not transformed. This means that they will probably be more accurate in predicting high values than low values. Consequently, transformations such as log-transform will reduce the relative importance of these high values, hence generalizing better.

In the context of the present study, the whole purpose of data transformation was to reduce the impacts of the extremely high values on model calibration. This is because those high values often present in extremely low proportions within the data. If those extreme values were left untransformed, they may cause the models to emphasize too much on rare extreme events, and thus largely affect our ability to represent the overall large-scale patterns in water quality. Our transformed model focuses more on proportional errors instead of absolute errors since the latter is less important at high concentrations in practice. Additionally, we presented model evaluation at the transformed scale because the model was calibrated in a transformed scale, and we believe that the transformed scale is most relevant and informative for performance assessments as we presented.

To address this comment, we will add more discussions in Section 2.2 (Data collection and processing) to improve the justification of data transformation. We will also add explanations in Section 4.1 (Model performance assessment) on why model performance evaluations are presented in a transformed scale. While we decided to focus this study on the transformed data, we have noted back-transforming modelled data as a possible option, and we would like to explore the differences between these two approaches in future studies.

*I would like to see comparison of this model to benchmark models. For example, how does this compare against a simple linear regression? Against a mixed effects regression? Against simply fitting linear trends independently at each site? Not all of these need to be done, but some sort of well selected benchmarks are needed to contextualize model performance.*

There has been extensive theoretical exploration and testing of random forest methods against other methods. By way of example, performance of the random forest algorithm in simulating

groundwater nitrate contamination at the African continent scale was evaluated in comparison to the multiple linear regression by Ouedraogo et al. (2019). Moreover, Chen et al. (2020) compared the water quality prediction performance of several learning models, including logistic regression, linear discriminant analysis, support vector machine, decision tree, random forest, etc., across major rivers and lakes in China. In addition, there has been extensive use of other data-based methods for predicting water quality, especially at a catchment scale (please see Section 1.2 of the paper and references therein). However, the fact that these simpler methods do not appear in the global water quality literature can be taken as being indicative that they are not suitable for that purpose. Some of these methods can be eliminated a priori: the processes are clearly not linear, and sites are not independent, so we would expect any reviewer to dismiss adoption of these methods for such a complex task as being foolish. Hence, we believe this 'model comparison' is beyond the scope of current study.

*How to performance metrics compare with similar nitrogen modeling studies? If this model is the core advancement of the paper, its performance relative to other literature has to be clear and impressive. Unclear at this point if that is the case.*

Thanks for this comment. To authors' knowledge, there is no parallel in the literature with similar model building approach, in terms of time scale, spatial resolution, and the selected constituent. However, we will add some evaluation of the closest relevant studies in the revised manuscript to discuss the similarity and differences between our findings and that of others.

**Model interpretation:**
*The variable importance feature is interesting, but I want to see the influence of each variable on the outcome to check they make scientific sense. Otherwise the model could be getting it 'right' for the 'wrong' reasons. Partial dependence plots or the like are one way to plot those dependencies and could provide more interesting scientific findings rather than the surficial relationships presented so far in the paper.*

Thanks for this suggestion. We believe that the variable importance measure used in this study (i.e., random forest permutation accuracy importance) is statistically advantageous compared to partial dependence plots or other univariate screening methods because it accounts for multivariate interactions.

In fact, partial dependence plots show how the model's predictions are affected by one or two predictors, which is the marginal effect that one or two features have on the predicted outcome of a machine learning model. In contrast, permutation accuracy importance measure covers the impact of each predictor variable individually as well as in multivariate interactions with other predictor variables, which is the combined importance of the variable and all its interactions with

other variables (see, e.g., Lunetta et al., 2004; Hastie et al., 2009; Grömping, 2009; Touw et al., 2013; Boulesteix et al., 2015; etc.). Moreover, partial dependence plots are easy to interpret. In the revised manuscript, we will add these plots and the above-mentioned discussion on their advantages and/or disadvantages when dealing with high-dimensional problems.

Literature discussion:
***Overall it did not seem like the results were sufficiently contextualized in the literature.*** *This goes for the performance metrics and the identification of increasing/decreasing trends in certain regions. Several of these regions have already been identified as having increasing/decreasing trends so how do these results build off of (or contradict) the prior literature?*

We believe that our paper has been sufficiently contextualized and historicized, particularly in the Introduction section  by (i) placing our research topic within its larger setting, (ii) providing important perspective by citing similar examples or relevant background, (iii) explaining what historical circumstances led up to the topic we are discussing, (iv) citing other scholars who have recently contributed to the field, and (v) exploring how our analysis fits into a larger discussion about the field. Also, we agree that the regional observations would benefit from further contextualization in relation to previous regional studies, which we will do in the revised manuscript.

*Figure 3 is not helpful, perhaps move to SI if authors feel it is relevant.*

This figure schematically shows the structure of the proposed random forest model. We believe it can help readers who are less experienced with the method to better understand how our WQ model works. From this perspective, the authors feel comfortable that the current place of Fig. 3 is appropriate.

*Figures 4 and 5: What do the observed and predicted look like in original units? If this model and data outputs will ultimately be useful, it has to perform well in the original units. Figure 5 (test data) is more relevant than Figure 4 (training data), so Fig 4 could go to the SI.*

To resolve this comment, we will add a new figure showing model performance in original units (not in transformed space), while keeping Figure 4 in the main text. For detailed discussion on why we presented results in box-cox transformed space, please see our response to your previous comment regarding presenting results in log-log space.

*Figures 6-8 **I worry considerably about extrapolation, so I do not trust the majority of locations shown.
Also, how about accompanying uncertainty maps?***

We understand that the reviewer is concerned about the spatial overfitting of the model. As proposed in our response to the second comment regarding extrapolation issue, apart from the conventional out-of-sample validation strategy discussed in the paper (Fig. 5) where the datasets which does not belong to the 1992-2010 period was used for out-of-sample testing, we will implement a 'target-oriented' validation strategy to address this concern in the revised manuscript. We are currently running Leave-Time-and-Location-Out Cross-Validation (LLTOCV) process, and we will update the relevant results and discussions in the revised manuscript.

Regarding the uncertainty analysis, we will add a new 'Discussion' section to better explain caveats/limitations of our approach and will thoroughly discuss the possible sources of uncertainty in our study.

*Figure 8: Adds little not shown elsewhere.*

Thanks for this comment. The purpose of this demonstration of latitudinal distribution of average (annual) estimated NOx—N concentrations is twofold. First, it helps us characterize the dominant regions of nitrogen concentrations. In addition, this figure shows that, in specific regions, high nitrogen fertilizer use does not necessarily correspond with equally high nitrogen concentration (e.g., the upstream region of the Yangtze River, upstream region of the Yellow River (China), upstream region of the Mississippi River (U.S.A), Murray River (Australia), Nelson River (Canada), midstream region of the Amur River (Russia, China), etc.). On the other hand, as can be seen from this figure there are some regions with relatively low nitrogen fertilizer use but high nitrogen concentration (e.g., downstream region of the Amazon River, and the midstream region of Congo River). This clearly indicates that the processes of the nitrogen cycle are complex, and dynamics of in-stream nitrogen concentration is controlled by nitrogen load input, hydrometeorological conditions, and management practices. Second, this figure can be used to validate our random forest model based on previous works that reported similar findings in these latitudes which correspond to the high agricultural activity and high livestock densities (see, e.g., Potter and Ramankutty, 2009; Lu and Tian, 2017; He et al., 2011).  To address this comment, we will expand relevant discussions in Section 4.2 (Patterns of nitrogen concentrations and global hotspots).

*Figure 9: Why do they find time series is more relevant? Is that surprising?*

We are unclear on the interpretation of this comment, and don't understand why reviewer anticipates a "*surprise*". As we mentioned in the manuscript, time-related variables, namely Cumulative Month since 1992 (CM) and Month of the Year (MOY), have been included in our model to represent 'distance' in the time domain, particularly to capture the long-term trends (CM) and to model seasonality effects (MOY). Furthermore, CM somehow can compensate for biogeochemical legacy and the long travel time between N input and riverine N export signals. Variable importance results shown in Fig. 9 indicate that the most important covariate for predicting monthly nitrogen concentration given the utilized global datasets is: **time**, i.e., cumulative and/or month of the year. These covariates allow the random forest model to fit different spatial patterns for each month underpinning that the observed nitrogen level is different from month to month.

*Is it interesting that cattle is ranked where it is? The 'so what?' is missing here.*

Yes, we believe it is an important finding. The direct inclusion of the livestock-related variables in data-driven, empirical water quality models is not common. To our knowledge, there is no machine learning-based water quality model that explicitly accounts for various livestock categories at global scale. Although it has been often argued that livestock, and in particular cattle population, play a key role in water pollution, we could not identify any similar study to confirm this argument using model-based evidence. This explains why our research was needed and highlights the significance of our paper.

**References**

Archer, K. J., & Kimes, R. V. (2008). Empirical characterization of random forest variable importance measures. Computational statistics & data analysis, 52(4), 2249-2260.

Auret, L., & Aldrich, C. (2011). Empirical comparison of tree ensemble variable importance measures. Chemometrics and Intelligent Laboratory Systems, 105(2), 157-170.

Behrens, T., Schmidt, K., Viscarra Rossel, R. A., Gries, P., Scholten, T., & MacMillan, R. A. (2018). Spatial modelling with Euclidean distance fields and machine learning. European journal of soil science, 69(5), 757-770.

Beusen, A. H. W., Van Beek, L. P. H., Bouwman, A. F., Mogollón, J. M., and Middelburg, J. J.: Coupling global models for hydrology and nutrient loading to simulate nitrogen and phosphorus retention in surface water–description of IMAGE–GNM and analysis of performance, Geoscientific Model Development, 8(12), 4045-4067.

Boulesteix, A. L., Janitza, S., Hapfelmeier, A., Van Steen, K., & Strobl, C. (2015). On the term 'interaction'and related phrases in the literature on Random Forests. *Briefings in bioinformatics*, 16(2), 338-345.

Canion, A., McCloud, L., & Dobberfuhl, D. (2019). Predictive modeling of elevated groundwater nitrate in a karstic spring-contributing area using random forests and regression-kriging. Environmental Earth Sciences, 78(9), 1-11.

Chen, K., Chen, H., Zhou, C., Huang, Y., Qi, X., Shen, R., ... & Ren, H. (2020). Comparative analysis of surface water quality prediction performance and identification of key water parameters using different machine learning models based on big data. Water research, 171, 115454.

Grabowski, Z. J., Watson, E., & Chang, H. (2016). Using spatially explicit indicators to investigate watershed characteristics and stream temperature relationships. Science of the Total Environment, 551, 376-386.

Gregorutti, B., Michel, B., & Saint-Pierre, P. (2017). Correlation and variable importance in random forests. Statistics and Computing, 27(3), 659-678.

Grömping U. Variable importance assessment in regression: linear regression versus random forest. *Am Stat*. 2009; 63(4):308–19.

Hastie T, Tibshirani R, Friedman JJH. *The Elements of Statistical Learning*, 2nd edn. New York: Springer; 2009.

He, B., Kanae, S., Oki, T., Hirabayashi, Y., Yamashiki, Y., & Takara, K. (2011). Assessment of global nitrogen pollution in rivers using an integrated biogeochemical modeling framework. Water research, 45(8), 2573-2586.

He, C. S. and DeMarchi, C.: Modeling spatial distributions of point and nonpoint source pollution loadings in the Great Lakes Watersheds, International Journal of Environmental Science and Engineering, 2(1), 24-30, 2010.

Li, J., Heap, A. D., Potter, A., & Daniell, J. J. (2011). Application of machine learning methods to spatial interpolation of environmental variables. Environmental Modelling & Software, 26(12), 1647-1659.

Lintern, A., Webb, J. A., Ryu, D., Liu, S., Bende-Michl, U., Waters, D., ... & Western, A. W. (2018). Key factors influencing differences in stream water quality across space. Wiley Interdisciplinary Reviews: Water, 5(1), e1260.

Lu, C., & Tian, H. (2017). Global nitrogen and phosphorus fertilizer use for agriculture production in the past half century: shifted hot spots and nutrient imbalance. Earth System Science Data, 9(1), 181-192.

Lunetta KL, Hayward LB, Segal J, Eerdewegh PV: Screening Large-Scale Association Study Data: Exploiting Interactions Using Random Forests. *BMC Genetics* 2004, 5: 32.

Mayorga, E., Seitzinger, S. P., Harrison, J. A., Dumont, E., Beusen, A. H., Bouwman, A. F., ... and Van Drecht, G.: Global nutrient export from WaterSheds 2 (NEWS 2): model development and implementation, Environmental Modelling & Software, 25(7), 837-853.

Meng, X., Hand, J. L., Schichtel, B. A., & Liu, Y. (2018). Space-time trends of PM2. 5 constituents in the conterminous United States estimated by a machine learning approach, 2005–2015. Environment international, 121, 1137-1147.

Meyer, H., Reudenbach, C., Hengl, T., Katurji, M., & Nauss, T. (2018). Improving performance of spatio-temporal machine learning models using forward feature selection and target-oriented validation. *Environmental Modelling & Software*, 101, 1-9.

Nicodemus, K. K. (2011). On the stability and ranking of predictors from random forest variable importance measures. Briefings in bioinformatics, 12(4), 369-373.

Nicodemus, K. K., & Malley, J. D. (2009). Predictor correlation impacts machine learning algorithms: implications for genomic studies. Bioinformatics, 25(15), 1884-1890.

Ouedraogo, I., Defourny, P., & Vanclooster, M. (2019). Application of random forest regression and comparison of its performance to multiple linear regression in modeling groundwater nitrate concentration at the African continent scale. Hydrogeology Journal, 27(3), 1081-1098.

Peterson, E. E., Sheldon, F., Darnell, R., Bunn, S. E., & Harch, B. D. (2011). A comparison of spatially explicit landscape representation methods and their relationship to stream condition. Freshwater Biology, 56(3), 590-610.

Potter, P., Ramankutty, N., Bennett, E. M., & Donner, S. D. (2010). Characterizing the spatial patterns of global fertilizer application and manure production. Earth interactions, 14(2), 1-22.

Saha, A., Basu, S., & Datta, A. (2021). Random forests for spatially dependent data. Journal of the American Statistical Association, 1-19.

Staponites, L. R., Barták, V., Bílý, M., & Simon, O. P. (2019). Performance of landscape composition metrics for predicting water quality in headwater catchments. Scientific reports, 9(1), 1-10.

Strobl, C., Boulesteix, A. L., Kneib, T., Augustin, T., & Zeileis, A. (2008). Conditional variable importance for random forests. BMC bioinformatics, 9(1), 1-11.

Toloşi, L., & Lengauer, T. (2011). Classification with correlated features: unreliability of feature ranking and solutions. Bioinformatics, 27(14), 1986-1994.

Touw WG, Bayjanov JR, Overmars L, Backus L, Boekhorst J, Wels M, van Hijum SAFT. Data mining in the life sciences with random forest: a walk in the park or lost in the jungle? *Brief Bioinform*. 2013; 14(3):315–26.

Wei, J., Huang, W., Li, Z., Xue, W., Peng, Y., Sun, L., & Cribb, M. (2019). Estimating 1-km-resolution PM2. 5 concentrations across China using the space-time random forest approach. Remote Sensing of Environment, 231, 111221.

---

## Author Comment (AC2)

**Response to Reviewer's Comments**

This document contains copies of all comments of the reviewer 2 (*in italicized, blue* text) and our planned effort to address them (in normal, black text). Our proposed manuscript revisions are underlined.

**Reviewer 2**

*The manuscript, titled "A global assessment of nitrogen concentrations using spatiotemporal random forests" by Sheikholeslami and Hall, introduced a machine learning (ML) approach (random forest model - RF) for predicting in-stream nitrogen (NOx-N) concentrations at the global scale. According to the authors, the novelties of this work are (1) its global scale application and (2) the spatio-temporal RF approach proposed in this study. In general, the manuscript was well written. Despite the results (instream NOx-N concentration) look quite well, the are several points regarding the model/approach used in this study that need to be addressed.*

Thanks for reviewing our manuscript and providing extensive and valuable suggestions. The constructive comments of this reviewer are highly appreciated. We will address all the comments as described in this rebuttal document.

General comments:
***1) Representation of nitrogen (N) lag times from input to riverine N export****: For water quality (e.g., N) modeling, it is expected that there could be significant N accumulated in the soil as biogeochemical legacy and the long travel time within the unsaturated/groundwater zone that could result in a lag time of years to decades between N input and riverine N export signals (e.g., Meals et al., 2010; Van Meter et al., 2017; Chen et al., 2018). It is unclear to me how the proposed RF model could take into account these factors. From my interpretation of the result, the "cumulative month count" variable (Figure 9) somehow could compensate for this kind of effect. However, we should not try to get the right result for the wrong reason.*

Thank you for sharing this important point and for providing the useful references. We certainly agree with reviewer's comment that legacy nutrients stored in soils (and groundwater) can create time lags between changes in nutrient inputs and the response of nutrient outputs. We also confirm reviewer's interpretation that the 'cumulative month of the year' somehow might compensate for legacy effect. An alternative approach would be using other machine learning algorithms such as hybrid long short-term memory-random forest method as proposed by Ahmed et al. (2021). This multi-model deep learning algorithm enables adding memories and lags of the predictors into the model. We will be very happy if other suggestions for how to address this challenge in machine learning-based water quality modelling become the focus of future discussion in the community. In the revised manuscript, we will provide reflections on this critical issue that have to be addressed in future studies.

Nonetheless, we should highlight that there is evidence that the diversity of nutrient recovery trajectories following reductions in nutrient loading suggest that nutrient removal capacity, contemporary nutrient loading, and nutrient-specific transport dynamics are as or more important than nutrient legacy in determining nutrient fluxes in watersheds, particularly when considering anthropogenic factors at large scales (see, e.g., Frei et al., 2021; Frei et al., 2020; Abbott et al., 2018; etc.). We will also elaborate more on this with the supporting literature.

*__2) Variable importance:__ Why are the month of the year and the cumulative month count the most important variables?*

Note that these variables were included in our model to represent 'distance' in the time domain, particularly to capture the temporal dynamics of nitrogen concentrations. Variable importance results shown in Fig. 9 indicate that the most important covariate for predicting monthly nitrogen concentration given the utilized datasets is: time, i.e., cumulative (CM) and/or month of the year (MOY). This reveals that seasonality effect and long-term trends play an important role in prediction accuracy of the random forest algorithm. To address this comment, we will add more details to the relevant discussions in Section 4.3 (High importance factors influencing predictions of nitrogen levels).

*I am wondering if the data used in the model has a strong seasonality that makes the variable "month of year" really matter. If this is the case, what is the implication for model application/performance in other areas that have less/no clear seasonality?*

This is an excellent concern. Yes, apparently the input data of the model has a considerable seasonality that makes the variable "month of year" important. Our finding is justifiable noting that generally there is a seasonal variability in major factors influencing water quality, such as vegetation, land-use change, hydro-climatic parameters, and farming activities, which strongly influence constituents' concentrations in different seasons (see, e.g., Pejman et al., 2009; Shabalala et al., 2013; Xu et al., 2019; etc.). Therefore, it is expected to observe seasonal variation of water quality in many regions of the world. In the revised manuscript, we will add more discussion in Section 4.3 to highlight seasonal changes in water quality and its main influencing factors.

Regarding the implication for model application in areas that have less/no clear seasonality, we anticipate the proposed model can learn from given data that the month of the year is relevant or not. This is the fundamental ability of the learning algorithms that they can uncover and extract useful information from the input data (e.g., spatial and/or temporal variations).

*Is the predictor "cumulative month count" highly important because of an increasing trend in the output variables in many areas (lines 495-497)? If yes, what are the implications from this?*

First, we have to clarify that the importance of this variable measured by random forest does not necessarily imply that there is a strong increasing/decreasing trend in target variable (i.e., nitrogen level). In fact, random forest evaluates variable importance by estimating the mean decrease in prediction accuracy before and after randomly permuting the values of a given predictor. Therefore, as we mentioned in our response to your second comment, our factor importance results indicate that the most important covariate for predicting monthly nitrogen concentration given the utilized datasets is: *time*, i.e., cumulative (CM) and/or month of the year (MOY). In other words, the seasonality effect, and long-term trends play an important role in prediction accuracy of the random forest model. But, one cannot say if there is an increasing or decreasing trend in nitrogen level merely based on factor importance results.

*Why does "fertilizer application" have a low rank?*

We disagree with the reviewer's interpretation that "fertilizer application" is among the less-influential factors. In contrast, as mentioned in Section 4.3, nitrogen fertilizer use is one of the most important factors. In fact, as shown in Fig. 9, the strongly influential predictors are (in rank order): (i) cattle population, (ii) nitrogen fertilizer use, (iii) temperature, (iv) precipitation.

*Why is there not much difference in the variables that were ranked 3rd to 15th (Figure 9)?*

We believe that for the top 7 important variables the ranking is distinguishable. However, for the rest of predictors, i.e., 8th to 15th, we agree that the difference in variable importance values is not significant. It means that randomly permuting the values of these predictors resulted in quite the same change in prediction accuracy. In other words, the importance of these variables cannot be robustly ranked, even though they are all influential. As mentioned in the manuscript (Section 5), to comprehensively analyze how various factors influence model output variability, a more advanced approach is required. Global sensitivity analysis methods are suitable candidates in this regard.

*3) Spatial unit: For predicting instream nitrogen concentrations, it is not clear to me why the authors did not use river network (instead of grid cell) as a spatial unit. In 1 grid cell (size of ≈ 55 km2) there could be*

*several rivers, so it is unclear if the predicted values are applied for the main or tributary rivers. In addition, with rivers in big basins (e.g., Elbe, Mississippi, Amazon, Mekong River Basins, etc) that are running across multiple grid cells, it would be useful to incorporate the effect of upstream management/catchment characteristics into the model, it is not clear if this was considered in the RF model or not. As I understood from the description, the predictors for instream N concentration prediction currently only cover the properties within the grid cell of interest (no consideration of information from the upstream grid cell for large rivers).*

Thank you, this is a very good point, and we confirm that your interpretation about the spatial resolution of our study is all correct, i.e., the current covariates used for predicting in-stream nitrogen concentrations only cover the properties within the grid cell of interest.

It is widely understood that human activities modify runoff regimes in different spatio-temporal scales and has been proven by several studies, according to long-term observations and/or hydrological modelling experiments (see, e.g., Ren et al., 2002; Zhang and Schilling et al., 2006; Ferguson and Maxwell, 2012). Therefore, we think the impacts of upstream management and human interventions (e.g., land use changes, reservoir operation, river network modification, and building of dams) were, at least implicitly, reflected in runoff data as one of the predictor variables. We'd like to explore the impacts of other human-induced modifications on nitrogen concentrations by inclusion of other variables directly into the model, such as groundwater extraction, irrigation withdrawals, and existence of dams, in future improvements.

Regarding the catchment characteristics, this can be resolved, for example, by adding hydrography data delineating global river networks, though it will presumably add more complexity to the model. In the revision, we will add more variables to the list of predictors, including upstream characteristics, stream proximity (e.g., distance up to the stream) or log-transformed flow accumulation for better capturing spatial characteristics of watersheds. Previous studies have shown that these variables can be key drivers of water quality responses in rivers (see, e.g., Staponites et al., 2017; Lintern et al., 2017; Grabowski et al., 2016; Peterson et al., 2010). Particularly, they reported that accounting for the hydrological flow paths and flow accumulation through the landscape and coupling these processes with specific landscape features can improve model performance. We will also elaborate more on this with the supporting literature.

***4) Capacities/limitation section****: I would suggest including a section describing the model capacities and limitations of the model/approach. Although the authors mentioned these points briefly in the conclusion section, however, they could be extended in a separate section.*

We highly appreciate this very important comment. In the revised manuscript, we will add a new Discussion section to better explain caveats/limitations of our approach and will provide possible recommendations for future research.

Specific comments:
*Lines 30: "In addition, extensive construction of dams, excessive extraction of groundwater, deforestation, and expanding agricultural land use have altered sedimentary processes, mobilization of salts, and nutrient export to river systems, all of which drive WQ deterioration and groundwater pollution in many parts of the world…". Were these factors considered in the model?*

Our model only accounts for expanding agricultural land use in terms of 'cropland area' and 'fertilizer use'. In the revised manuscript, we will also add 'forest fraction' and 'urban fraction' to the list of predictor variables, which will partially address this concern. We think addition of these variables can help us better capture the impact of anthropogenic forces on the global nitrogen variability in surface waters. Moreover, we'd like to explore the impact of other factors such as groundwater extraction and the existence of dams in future model improvements.

*Lines 183-184: What is the temporal resolution of the NOxâ-N data and how were they aggregated to monthly timestep?*

Most of the GEMStat stations provide monthly measurements. When daily values were reported for a specific month, we simply used the median of the daily measurements for that specific month. We will revise the text (Section 2.1) and add more details on this during revision.

*Line 206: Please indicate where the readers could find the list of 27 potential explanatory variables*

As mentioned in the paper, the selection of these predictor variables was based a process-informed manner through an extensive literature review (please see Section 2.2 and the references therein). To address this comment, we will also provide the list of 27 potential explanatory variables in Appendix of the revised manuscript.

*Line 287: "The second strategy …" The second strategy has not been mentioned before this point*

As mentioned in line 287, the second strategy is using geographic information as an auxiliary input to capture the spatial variability of the target variable, for instance, by adding geographic

coordinates (Behrens et al., 2018; Meng et al., 2018) or other spatial distances (Li et al., 2011; Wei et al., 2019) into the list of predictors. We will revise this paragraph to improve the clarity of the writing.

*Line 291: "Cumulative Month since 1992": how sensitive are the results to the start of the month count? This is a critical point if someone wants to run the model for other periods*

Thank you, this is a very good point. Note that this variable has been included in our model to capture the long-term trends. Therefore, this variable can be critical when running our model for other periods that are much longer than our study period (1992-2010) and when the start of the time is far from 1992 as well. We will also discuss this point in the Results to highlight the importance of this factor.

*Line 295: "..17 variables" – Please point out the list of 17 variables*

Thank you, we will address this comment in the revised manuscript.

*Figure 3: I would suggest adding more frames to the "output" panel (as already done in the "input" panel) to reflect that the spatial and temporal properties of outputs*

Good point. In the revised manuscript, we will modify Fig. 3.

*Lines 418-419: Is there a high correlation between elevation and latitude/longitude?*

As mentioned in the manuscript, we have used DEM along with latitude and longitude as predictor variables in our model. Simply speaking, DEM is a discrete representation of the surface of the Earth using points generally placed on a regular grid. For each point its position is known in a chosen reference frame and represented through a chosen coordinate system (horizontal coordinates: geographic (latitude, longitude) or cartographic (East, North); elevation (heights or depths): orthometric with respect to a chosen geoid model or ellipsoidal. In other words, data sets can be either UTM-based, with points on a regular rectangular grid, or lat/long-based, with points at regular intervals on the geographic grid. Because of earth curvature, the two types of data behave differently over large areas. We have not conducted a correlation analysis to investigate the relationship between elevation and lat/lon. We believe this issue is beyond the scope of current study.

*Lines 426 – 430: back to table 1, it is not clear whether these data (livestock population) were assumed to be time-invariant or time-variant*

For our analysis, we assumed livestock population and wastewater production are constant for the study period mainly due to lack proper information regarding temporal variability of them. We will explain this and revise Table 1 to describe which predictors are time-variant/invariant.

*Line 447-448: "This might be partially due to a high correlation between the agricultural fraction of land area and nitrogen fertilizer use" – Why do highly correlated variables have different rankings?*

As we mentioned in the manuscript, several studies showed that estimating the importance of predictors by random forest algorithm might be problematic in the presence of correlation between variables. As another example, Toloşi and Lengauer (2011) identified the same issue based on extensive numerical experiments, which they called it "correlation bias". In summary, most of these studies reported two key effects of the correlation on the permutation importance measure: (1) the importance values of the most discriminant correlated variables are not necessarily higher than a less discriminant one, and (2) the permutation importance measure depends on the size of the correlated groups.

Gregorutti et al. (2017) provided theoretical validations for this issue, in a particular statistical framework. Based on Gregorutti et al. (2017)'s study, one possible answer for the reviewer's question on 'why correlated variables exhibit distinct ranking when using permutation-based importance measure', is that when one of the two correlated variables is permuted, the error does not increase that much because of the presence of the other variable, which carries a similar information. The value of the prediction error after permutation is then close to the value of the prediction error without permutation and the importance is small. In addition, it has been asserted that the permutation-based importance measure typically tends to discard most of the correlated variables even if they are discriminants and randomly selects one representative among a set of correlated predictors as the most influential factor (Gregorutti et al., 2017; Bühlmann et al., 2013). We will strengthen relevant discussions on the correlation and variable importance in random forests to better justify our results in Section 4.3 and provide more appropriate references.

*Table 2: How were annual time step data (especially fertilizer application) disaggregated to monthly time step?*

We assumed that these predictors, i.e., population, cropland area, and synthetic nitrogen fertilizer use, are invariant in month (i.e., only the annual variability is considered), and thus no

disaggregation was performed. In other words, for each month of the year, we used the annual value of that year. We will modify Table 1 and revise Section 2.2 to clarify this.

*Table 1: Please provide full names for the technical terms (e.g., ANN, DT, MLT,…) in Table 1*

We will correct this as suggested.

**References**

Abbott, B. W., Moatar, F., Gauthier, O., Fovet, O., Antoine, V., & Ragueneau, O. (2018). Trends and seasonality of river nutrients in agricultural catchments: 18 years of weekly citizen science in France. Science of the Total Environment, 624, 845-858.

Ahmed, A. M., Deo, R. C., Feng, Q., Ghahramani, A., Raj, N., Yin, Z., & Yang, L. (2021). Deep learning hybrid model with Boruta-Random forest 8ptimizer algorithm for streamflow forecasting with climate mode indices, rainfall, and periodicity. Journal of Hydrology, 599, 126350.

Behrens, T., Schmidt, K., Viscarra Rossel, R. A., Gries, P., Scholten, T., & MacMillan, R. A. (2018). Spatial modelling with Euclidean distance fields and machine learning. European journal of soil science, 69(5), 757-770.

Bühlmann, P., Rütimann, P., van de Geer, S., & Zhang, C. H. (2013). Correlated variables in regression: clustering and sparse estimation. Journal of Statistical Planning and Inference, 143(11), 1835-1858.

Ferguson, I. M., & Maxwell, R. M. (2012). Human impacts on terrestrial hydrology: climate change versus pumping and irrigation. Environmental Research Letters, 7(4), 044022.

Frei, R. J., Abbott, B. W., Dupas, R., Gu, S., Gruau, G., Thomas, Z., ... & Pinay, G. (2020). Predicting nutrient incontinence in the Anthropocene at watershed scales. Frontiers in Environmental Science, 7, 200.

Frei, R. J., Lawson, G. M., Norris, A. J., Cano, G., Vargas, M. C., Kujanpää, E., ... & Abbott, B. W. (2021). Limited progress in nutrient pollution in the US caused by spatially persistent nutrient sources. PloS one, 16(11), e0258952.

Grabowski, Z. J., Watson, E., & Chang, H. (2016). Using spatially explicit indicators to investigate watershed characteristics and stream temperature relationships. Science of the Total Environment, 551, 376-386.

Gregorutti, B., Michel, B., & Saint-Pierre, P. (2017). Correlation and variable importance in random forests. Statistics and Computing, 27(3), 659-678.

Li, J., Heap, A. D., Potter, A., & Daniell, J. J. (2011). Application of machine learning methods to spatial interpolation of environmental variables. Environmental Modelling & Software, 26(12), 1647-1659.

Lintern, A., Webb, J. A., Ryu, D., Liu, S., Bende-Michl, U., Waters, D., ... & Western, A. W. (2018). Key factors influencing differences in stream water quality across space. Wiley Interdisciplinary Reviews: Water, 5(1), e1260.

Pejman, A. H., Bidhendi, G. R., Karbassi, A. R., Mehrdadi, N., & Bidhendi, M. E. (2009). Evaluation of spatial and seasonal variations in surface water quality using multivariate statistical techniques. International Journal of Environmental Science & Technology, 6(3), 467-476.

Peterson, E. E., Sheldon, F., Darnell, R., Bunn, S. E., & Harch, B. D. (2011). A comparison of spatially explicit landscape representation methods and their relationship to stream condition. Freshwater Biology, 56(3), 590-610.

Ren, L., Wang, M., Li, C., & Zhang, W. (2002). Impacts of human activity on river runoff in the northern area of China. Journal of Hydrology, 261(1-4), 204-217.

Shabalala, A. N., Combrinck, L., & McCrindle, R. (2013). Effect of farming activities on seasonal variation of water quality of Bonsma Dam, KwaZulu-Natal. South African Journal of Science, 109(7), 1-7.

Staponites, L. R., Barták, V., Bílý, M., & Simon, O. P. (2019). Performance of landscape composition metrics for predicting water quality in headwater catchments. Scientific reports, 9(1), 1-10.

Toloşi, L., & Lengauer, T. (2011). Classification with correlated features: unreliability of feature ranking and solutions. Bioinformatics, 27(14), 1986-1994.

Wei, J., Huang, W., Li, Z., Xue, W., Peng, Y., Sun, L., & Cribb, M. (2019). Estimating 1-km-resolution PM2. 5 concentrations across China using the space-time random forest approach. Remote Sensing of Environment, 231, 111221.

Xu, G., Li, P., Lu, K., Tantai, Z., Zhang, J., Ren, Z., ... & Cheng, Y. (2019). Seasonal changes in water quality and its main influencing factors in the Dan River basin. Catena, 173, 131-140.

Zhang, Y. K., & Schilling, K. E. (2006). Increasing streamflow and baseflow in Mississippi River since the 1940 s: Effect of land use change. Journal of Hydrology, 324(1-4), 412-422.

---

## Author Comment (AC3)

**Response to Reviewer's Comments**

This document contains copies of all comments of the reviewer 3 (*in italicized, blue* text) and our planned effort to address them (in normal, black text). Our proposed manuscript revisions are underlined.

**Reviewer 3**

*Sheikholeslami and Hall, 2022 utilize machine learning approach to build a random forest model of nitrogen concentration in major river basins. They apply the model to global river basins to identify nitrogen concentration hot-spots and decadal increase in nitrogen concentration. The manuscript applies random forest approach for estimating riverine nitrogen concentration, but their findings are not filling a gap in the literature. Their finding that highest nitrogen concentrations are observed in United States, Europe, China, and India is not new given that these regions are agriculture dominated and utilize large amount of fertilizer and manure. The manuscript lacks a novel motivation, several important predictor variables are not considered, and the final analysis does not provide any new information.*

Thank you very much for your comprehensive review and identification of key areas of improvement. We provide detailed response to your comments in the subsequent section. Below, we list your main concerns, and a summary of our proposed revisions to address them.

Overall, we believe that the reviewer has under-estimated the contribution in the paper. As outlined in this rebuttal document, the application of the random forest model can be considered to be state-of-the-art in a rapidly evolving field. The use of these techniques in the very challenging field of global water quality modelling is novel and has yielded predictive results that exceed other approaches in important respects. We think reviewer does not seem to have recognized (or has dismissed) the merit of the utilized methodology considering our comprehensive exercise in data processing, model selection, adaptation, and validation. We appreciate reviewer's suggestions on potential topics, and we provide responses to individual ones as following:

> **(1) Lack of a novel motivation**: First, despite a plethora of powerful machine learning methods, ANN is the most popular method for water quality modelling. To fill this gap, we implemented random forest algorithm to examine its predictive performance when applied to large spatial scales. Second, our thorough search of the relevant literature indicated that machine learning methods, particularly random forests, have not been implemented at global scale to identify nitrogen hotspots and key drivers of nitrogen variability. Third, despite being successful in simulating and predicting surface water quality at catchment-scale, machine learning methods have not been utilized to provide spatially explicit (gridded) estimates of nitrogen levels. Based on our observation,

almost all machine learning models are lumped in space (see Table 1). This gap further motivated our development of the spatiotemporal random forest-based global model in the present study. Please note that these gaps have been discussed in Section 1.2. In the revised manuscript, we will improve the writing to reduce the reviewer's confusion that appear to have arisen due to lack of sufficient clarity in the original versions.

**(2) Missing several important predictors**: It would be helpful to have an indicator from the reviewer what important predictors they consider to have been missed out. As proposed in this rebuttal document, based on the reviewers' suggestion we are also planning to include additional variables, such as forest fraction, urban fraction, and hydrography data, to the structure of the model. We will also investigate their impact on model performance during the revision.

**(3) Lack of new information**: First, we restate that the model provided much higher spatial resolution than we were able to contain in our high-level summary of the results. Our model was used to predict nitrogen concentrations in large river basins globally, providing new information about dynamics of nitrogen concentrations in location with scarce/no observations. Furthermore, we provide NOx-N concentrations (mg/l) worldwide (180°E–180°W; 90°S–90°N) at a spatial grain of 0.5-degree. The NOx-N concentrations, mapped across the globe for 1992-2010, are available in a compressed GeoTiff file format in the WGS84 coordinate reference system (EPSG:4326 code). The developed stream nitrogen concentration maps have a wide array of potential applications in stream ecology, biodiversity research, conservation science, and stream and lake restoration ecology. For instance, the produced maps can be used to quantify the overall mass of nitrogen discharged into a specific lake or ocean body, enabling a deeper understanding of global-scale eutrophication. Furthermore, our estimates of nitrogen concentration can be used to verify new process-based models that predict nitrogen concentrations and transformations in inland waters worldwide. We encourage potential users of the described geo-dataset to contact the authors for future product updates. We will add this to the revised manuscript to better highlight the usefulness of our findings.

In addition, to improve the linkage between study objectives and results, we propose to present additional results to highlight several model capabilities. These include model validation using a new validation strategy, adding partial dependence plots, presenting the distribution of R2 values, and further discussions on limitations of our approach along with recommendations for future studies.

Introduction:

*1. The authors include "three critical observations" from the literature review of ML applications for studying water quality. What is the reason for including these observations?*

As stated in our response to previous comment, these three observations essentially motivated us to develop the proposed random forest-based global water quality model. To resolve this comment, we will make necessary changes in the revised manuscript to improve the quality and presentation of the Introduction section.

*2. The two main objectives of the study related to identifying global nitrogen pollution hotspots and key drivers of nitrogen variability at a global scale are weak since various studies have examined these and it is not a novelty.*

Politely disagree. To authors' knowledge, there is no parallel in the literature in terms of time scale and spatial resolution with similar approach to identify nitrogen pollution hotspots and key drivers of nitrogen variability at a global scale. In fact, one essential purpose of the developed model is to enable water quality assessment and quantification of future scenarios (e.g., of livestock pasture extent) at a 'global scale'. Though some instances of analysis of these questions exist (see, e.g., Mayorga et al., 2010; He et al., 2011; Beusen et al., 2016; etc.) they all have inevitable limitations, and none have used our proposed approach which we believe yields worthwhile results. Therefore, we believe that both objectives are strong enough to motivate our analyses. The reviewer argued that "*various studies have examined these*" we'd be more than happy if the reviewer helps us identify these studies, so we can cite them in the revised manuscript. For more discussion, please see our response to your first comment in this rebuttal document.

*3. The Introduction currently has three section and is quite long. Condensing this section will improve the readability of the manuscript.*

We will condense this section in the revised version, while keeping some important background information for the broad readership of the HESS as a multi-disciplinary journal.

Data and Methodology:

*4. Lines 218-222: The authors mention reshaping all predictors to 0.5-degree resolution, but they have not described how precipitation, runoff, and other predictors from the entire upstream catchment/watershed is accounted for in predicting river nitrogen concentration. It appears that they only considered the contribution of the various predictors from the 0.5-degree grid. It is highly likely that only a fraction of this*

*grid falls in the upstream catchment and for several observations large fraction of the upstream catchment is not included in this grid. Thus, the estimation of contribution from various predictor variables is likely flawed.*

Thank you, for this important comment, and we confirm that your interpretation about the spatial resolution of our study is all correct, i.e., the current variable used for predicting in-stream nitrogen concentrations only cover the properties within the grid cell of interest.

Regarding the catchment characteristics, this can be resolved, for example, by adding hydrography data delineating global river networks, though it will presumably add more complexity to the model. In the revision, we will add more variables to the list of predictors, including upstream characteristics, stream proximity (e.g., distance up to the stream) or log-transformed flow accumulation for better capturing spatial characteristics of watersheds. Previous studies have shown that these variables can be key drivers of water quality responses in rivers (see, e.g., Staponites et al., 2017; Lintern et al., 2017; Grabowski et al., 2016; Peterson et al., 2010). Particularly, they reported that accounting for the hydrological flow paths and flow accumulation through the landscape and coupling these processes with specific landscape features can improve model performance. We will also elaborate more on this with the supporting literature.

*5. Adding latitude and longitude as predictor variables does not fully capture the spatial relationship between observations and predictors.*

Agreed. But, as we extensively discussed in Section 3.1.1, two strategies have been proposed in the literature to account for spatial information: (1) using a hybrid modelling framework by embedding Kriging and Gaussian process modelling into the standard random forest method (Saha et al., 2021; Canion et al., 2019); and (2) using geographic information as the auxiliary inputs, for example, adding geographic coordinates (Behrens et al., 2018; Meng et al., 2018) or other spatial distances (Li et al., 2011; Wei et al., 2019) into the list of predictors. Since adding lat/lon as predictor variables to represent spatial relationship is common in data-driven modelling of Earth and environmental systems, we followed this strategy and incorporated lat/lon into our random forest model. During the revision, we will highlight the weakness of the second strategy in not fully capturing the spatial relationships between observations and predictors.

*6. Additional land use predictors should be considered in the study for example, forest fraction, urban fraction.*

Thank you for this suggestion. Please note that from our review of literature we observe that cropland fraction is one of the chief determinants of the nitrogen variability, as evident from our variable importance analysis (Fig 9). To address this comment, additional predictors such as forest fraction and urban fraction will be added to the list of predictors. Further, we will examine the relative importance of these variables when simulating nitrogen concentrations.

*7. In addition to considering monthly precipitation, extreme precipitation variables and variables capturing dry spells should also be considered as they impact nitrogen concentration. For example, a long dry spell will result in high nitrogen concentrations however, if the dry spell is followed by a large precipitation event concentrations will drop. The monthly mean precipitation will ignore this temporal variability and thus not accurately predict nitrogen concentration.*

Thanks for sharing the interesting idea. Perhaps you missed it, but we have already discussed the impact of extreme and/or prolonged hydroclimatic events on nutrient concentrations in Section 4.3 (line 454-464). We believe quantifying the impact of extreme events on nitrogen concentrations using machine learning is an interesting topic, which we'd like to explore in future. In the Discussion section of the revised manuscript, we will mention this as an important recommendation for future research.

*8. Table 2 should be replaced with a table containing the final 17 predictor variables. In its current format the table 2 does not list the four time and space predictors and lists the livestock predictors in a single row that makes it appear as a single predictor not five different predictors.*

Thanks for this comment. We will update Table 2 in the revised manuscript based on reviewer's suggestion.

Results:
*9. In the Results section, the authors have not compared their findings with any of studies listed in Table 1. The authors should discuss the similarity and differences between their findings and that of others using similar model building approach.*

As we mentioned in our response to your second comment, there is no parallel in the literature with similar approach. None of the studies listed in Table 1 systematically investigated the key drivers of nitrogen variability, which was one of the main objectives of our study. Additionally, none of them provided spatially grided estimates of nitrogen concentration, as all of them were lumped in space. However, we will add some evaluation of the closest relevant studies in the revised manuscript to compare our findings and that of others.

We want to highlight that our paper has been sufficiently contextualized, particularly in the Introduction section by (i) placing our research topic within its larger setting, (ii) providing important perspective by citing similar examples or relevant background, (iii) explaining what historical circumstances led up to the topic we are discussing, (iv) citing other scholars who have recently contributed to the field, and (v) exploring how our analysis fits into a larger discussion about the field. Indeed, we agree that the regional observations would benefit from further contextualization in relation to previous regional studies, which we will do in the revised manuscript.

*10. In addition to the annual concentrations (Figure 6), authors should also analyze and discuss monthly or seasonal maximum concentration.*

Thanks for pointing this out. We will present more results on monthly/seasonal variability of maximum concentrations in the Results section of the revised manuscript.

*11. The authors had developed spatial plots of nitrogen concentration for every month between 1992-2010, then why did the limit the analysis of change in nitrogen concentration to a decadal scale difference only. For large river basins, they can perform trend analysis. This will me more useful information for policy makers than decadal scale difference.*

Thanks for this comment, we strongly believe that decadal analysis is also helpful to identify long-term trends. While we decided to focus this study on the decadal scale, we have noted trend analysis for the selected large river basins as a possible option, and we would like to explore its usefulness in future studies.

*12. Line 387 – 389: What factors contributed to the decline in nitrogen concentrations? Just merrily stating decline is not enough and the drivers behind this observation should be discussed especially given that few regions with highest nitrogen concentration (India and South Korea) also have the largest decline.*

Thank you for raising this point, and we confirm the reviewer's interpretation that significant decline in nitrogen concentrations has been occurred in a few basins with highest nitrogen concentrations. Our focus, however, was not to explore which factors were mainly responsible for decline in nitrogen level in some regions. We therefore think a thorough analysis of this question is beyond the scope of the present study. There might be several reasons for this observation, such as dietary behavior change, improved nitrogen fertilizer management, increasing efficiency of crop production, hydroclimatic regime shifts, change in upstream

management, etc. We will elaborate more on the possible causes of the observed decline in nitrogen concentrations with the supporting literature.

*13. The fact that month of year (MOY) is significantly more important than precipitation, runoff, and temperature seems concerning. If precipitation, runoff, and/or temperature, were selected as predictor variables it would have a direct physical meaning. For example, increase/decrease in precipitation can decrease/increase the nitrogen concentration.*

We are unclear on the interpretation of your comment. Please note that this variable has been included in our model to represent 'distance' in the time domain, particularly to capture the seasonality effect. Based on our factor importance analysis, therefore, the most important covariate for predicting monthly nitrogen concentration given the utilized datasets is: time. One possible reason is that the data used in the model has a strong seasonality that makes the variable "month of year" highly important. This can be justified because generally there is a considerable seasonal variability in major factors influencing nitrogen level, such as vegetation, land-use change, hydro-climatic parameters, and farming activities, which strongly influence constituents' concentrations in different seasons (see, e.g., Pejman et al., 2009; Shabalala et al., 2013; Xu et al., 2019; etc.). Consequently, it is expected to observe seasonal variation of water quality in many regions of the world. Furthermore, to avoid confusion, we have to clarify that the importance of this variable should not be miss-understood as a strong trend in the sense that the monthly nitrogen concentrations increase over time or the like.

*14. What is the physical meaning behind cumulative month (CM) variable being identified as the second most influential predictor?*

We should again highlight that this variable has been included in our model to capture the long-term trends. Furthermore, it somehow can compensate for biogeochemical legacy and the long travel time between N input and riverine N export signals. The covariate CM allows the random forest model to fit different spatial patterns for each month underpinning that the observed nitrogen level is different from month to month. To address this comment, we will add more details to the relevant discussions in Section 4.3 (High importance factors influencing predictions of nitrogen levels).

*15. Figure 9 - Why is the relative importance of 3-15 predictors almost same?*

We believe that for the top 7 important variables the ranking is distinguishable. However, for the rest of predictors, i.e., 8th to 15th, we agree that the difference in variable importance values is not significant. It means that randomly permuting the values of these predictors resulted in quite the same change in prediction accuracy. In other words, the importance of these variables cannot be robustly ranked, even though they are all influential. As mentioned in the manuscript (Section 5), to comprehensively analyze how various factors influence model output variability a more advanced approach is required. Global sensitivity analysis methods are suitable candidates in this regard.

**16. Did the most influential predictors vary over space and/or time?**

Of course. Considering the spatio-temporal variability in both target and predictor variables, we can assert that 'factor sensitivity' also varies over time and/or space. However, the inherent feature of the random forest algorithm for evaluating variables' predicting strength (expressed as variable importance ranking) can only measure variable importance for all input data using the whole space-time regression matrix. Thus, it cannot provide spatial-temporal characteristic of the factor sensitivity.

Assessing how the impact of predictors and their interactions varies in both space and time requires a more systematic approach, such as using advanced global sensitivity analysis methods, which we'd like to explore in future studies. To clarify this, we will add a brief discussion on spatio-temporal sensitivity analysis of random forest model and will elaborate more on this issue with the supporting literature.

**Reference**

Behrens, T., Schmidt, K., Viscarra Rossel, R. A., Gries, P., Scholten, T., & MacMillan, R. A. (2018). Spatial modelling with Euclidean distance fields and machine learning. European journal of soil science, 69(5), 757-770.

Beusen, A. H. W., Van Beek, L. P. H., Bouwman, A. F., Mogollón, J. M., and Middelburg, J. J.: Coupling global models for hydrology and nutrient loading to simulate nitrogen and phosphorus retention in surface water–description of IMAGE–GNM and analysis of performance, Geoscientific Model Development, 8(12), 4045-4067.

Canion, A., McCloud, L., & Dobberfuhl, D. (2019). Predictive modeling of elevated groundwater nitrate in a karstic spring-contributing area using random forests and regression-kriging. Environmental Earth Sciences, 78(9), 1-11.

He, B., Kanae, S., Oki, T., Hirabayashi, Y., Yamashiki, Y., & Takara, K. (2011). Assessment of global nitrogen pollution in rivers using an integrated biogeochemical modeling framework. Water research, 45(8), 2573-2586.

Li, J., Heap, A. D., Potter, A., & Daniell, J. J. (2011). Application of machine learning methods to spatial interpolation of environmental variables. Environmental Modelling & Software, 26(12), 1647-1659.

Mayorga, E., Seitzinger, S. P., Harrison, J. A., Dumont, E., Beusen, A. H., Bouwman, A. F., ... and Van Drecht, G.: Global nutrient export from WaterSheds 2 (NEWS 2): model development and implementation, Environmental Modelling & Software, 25(7), 837-853.

Meng, X., Hand, J. L., Schichtel, B. A., & Liu, Y. (2018). Space-time trends of PM2. 5 constituents in the conterminous United States estimated by a machine learning approach, 2005–2015. Environment international, 121, 1137-1147.

Pejman, A. H., Bidhendi, G. R., Karbassi, A. R., Mehrdadi, N., & Bidhendi, M. E. (2009). Evaluation of spatial and seasonal variations in surface water quality using multivariate statistical techniques. International Journal of Environmental Science & Technology, 6(3), 467-476.

Saha, A., Basu, S., & Datta, A. (2021). Random forests for spatially dependent data. Journal of the American Statistical Association, 1-19.

Shabalala, A. N., Combrinck, L., & McCrindle, R. (2013). Effect of farming activities on seasonal variation of water quality of Bonsma Dam, KwaZulu-Natal. South African Journal of Science, 109(7), 1-7.

Wei, J., Huang, W., Li, Z., Xue, W., Peng, Y., Sun, L., & Cribb, M. (2019). Estimating 1-km-resolution PM2. 5 concentrations across China using the space-time random forest approach. Remote Sensing of Environment, 231, 111221.

Xu, G., Li, P., Lu, K., Tantai, Z., Zhang, J., Ren, Z., ... & Cheng, Y. (2019). Seasonal changes in water quality and its main influencing factors in the Dan River basin. Catena, 173, 131-140.